# Harmonizing Gradient Matching For Fairness

**Ziwei Wu**                                            *ziweiwu2@illinois.edu*
*School of Information Sciences*
*University of Illinois Urbana-Champaign*

**Yikun Ban**                                          *banyikun2014@gmail.com*
*School of Information Sciences*
*University of Illinois Urbana-Champaign*

**Jingrui He**                                           *jingrui@illinois.edu*
*School of Information Sciences*
*University of Illinois Urbana-Champaign*

**Reviewed on OpenReview:** *https://openreview.net/forum?id=HMlK36YWWt*

## Abstract

Ensuring fairness across demographic groups is critical for machine learning systems deployed in high-stakes applications. Most existing approaches enforce fairness by directly minimizing disparities in predefined fairness metrics between groups, focusing primarily on the final model outcome. However, differences in distributions across groups can lead to heterogeneous optimization signals during training, resulting in imbalanced parameter updates and unstable fairness–performance trade-offs. In this work, we propose Fair Gradient Matching (FAIRGM), a fairness-aware optimization framework that harmonizes group-conditioned optimization signals. Instead of focusing solely on fairness metric disparities, FairGM aligns gradient signals of the fairness objective across groups at multiple levels of moments, including the zeroth-moment fairness metric itself, the first-moment mean gradients, and the second-moment gradient variances. These regularizations encourage similar optimization behavior across groups and lead to more stable fairness outcomes. To balance predictive performance and fairness objectives, we further formulate training as a multi-objective optimization problem and solve it using a Pareto-based optimization scheme. The resulting framework is compatible with a range of differentiable fairness metrics and gradient-based classifiers, supported by theoretical analysis connecting gradient alignment for fairness. Experiments on synthetic and real-world datasets demonstrate that FairGM achieves favorable fairness–accuracy trade-offs compared with existing fairness-aware learning methods, demonstrating the effectiveness and scalability of our approach.

## 1 Introduction

Machine learning models are increasingly deployed in high-stakes decision-making systems, including healthcare (Ahmad et al., 2018), criminal justice (Ghasemi et al., 2021), hiring (Mahmoud et al., 2019), and financial services (Acharya et al., 2024). In such applications, models that exhibit systematic disparities across demographic groups can lead to harmful societal consequences. Numerous studies have shown that modern machine learning models may produce predictions that differ significantly across groups defined by sensitive attributes such as race or gender, even when these attributes are not explicitly used as inputs (Hajian et al., 2016; Speicher et al., 2018). Building fair systems is thus of great significance in practice and has attracted considerable attention in recent years.

In response, a large body of work addresses this problem by enforcing parity in predefined fairness metrics across groups. Examples include equalized opportunity, equalized odds (Hardt et al., 2016), and accuracy

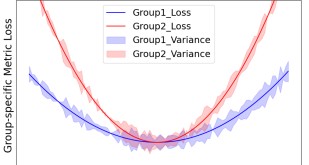 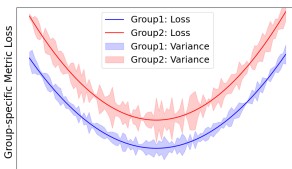 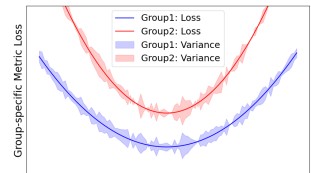 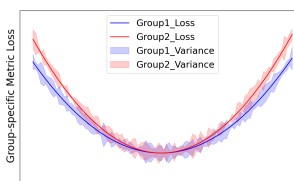

(a) Zeroth-moment gradient matching only.

(b) First-moment gradient matching only.

(c) Second-moment gradient matching only.

(d) Desired combined gradient matching.

Figure 1: Motivating examples of the combination of gradient matching under three moments. The x-axis represents the model parameter and the y-axis represents the group-specific fairness metric loss. The optimization of ERM is not guaranteed to always get to the exact optimal point in the fairness metric space. Thus, we are not only focusing on the optimal point but also pursuing a smooth neighborhood region around the optimal point. (a) is to simply ensure the fairness metric losses are close enough; (b) is to ensure the slopes of fairness metric trajectories are close enough; (c) is to ensure the variances of the fairness metric are close enough; (d) is our combined gradient matching method. Note that the lines of the two groups do not need to intersect at the lowest point. This figure only represents one case of the optimization landscape, which can be extended to more complicated loss landscapes.

parity (Zafar et al., 2017a). Many existing methods incorporate these notions through constraints or regularization terms that directly minimize disparities in the chosen fairness metric during training (Zafar et al., 2017b; Agarwal et al., 2018; Cotter et al., 2019; Maheshwari & Perrot, 2023). While these approaches can reduce fairness gaps at the final model solution, they typically focus only on the *outcome* of optimization rather than the *dynamics* of the optimization process itself.

However, the optimization dynamics of different groups can vary. In many real-world data sets, groups differ in their feature distributions, sample sizes, and noise characteristics. Consequently, gradient updates induced by different groups may point in different directions or exhibit different magnitudes during training. Such discrepancies in optimization signals can lead to imbalanced parameter updates, making it difficult to simultaneously achieve strong predictive performance and fairness across groups. Even when fairness metrics appear balanced at a particular parameter setting, the surrounding optimization landscape may remain asymmetric across groups. Since gradient-based optimizers do not converge to the exact optimum in practice, the final parameters lie in a neighborhood around it, and asymmetric gradient statistics in this neighborhood can reintroduce fairness disparities during training. These observations suggest that fairness-aware learning should not only align fairness metrics across groups, but also harmonize the *optimization signals* that different groups contribute during training. Gradients provide a natural representation of these signals (Charpiat et al., 2019), as they describe how model parameters should change to improve the fairness objective. Aligning gradients across groups therefore offers a promising mechanism for encouraging balanced optimization dynamics and improving fairness–accuracy trade-offs.

Motivated by this perspective, we propose Fair Gradient Matching (FairGM), a fairness-aware optimization framework that regularizes training by aligning group-conditioned gradients of a fairness metric. Our framework considers gradient statistics at multiple levels of granularity, as illustrated in Figure 1. First, we minimize the disparity of the *zeroth-moment fairness metric* itself across groups, which directly encourages metric-level parity. In Figure 1a, this corresponds to bringing the fairness losses of different groups closer in value. However, equalizing metric values alone does not guarantee stable optimization behavior: the gradients associated with the two groups may still differ significantly around the neighborhood of the optimum, meaning that small parameter updates can quickly reintroduce fairness disparities.

To address this, we further align the *first-moment mean gradients* of the fairness objective across groups. As illustrated in Figure 1b, matching mean gradients encourages the local slopes of the fairness loss landscapes to become similar across groups. This promotes parameter updates that move the model in directions that improve fairness for all groups simultaneously, rather than favoring one group over another. Nevertheless, aligning only the mean gradients may still overlook heterogeneity within each group, since individual samples can produce gradients with very different magnitudes or directions.

Therefore, we additionally align the *second-moment variance of gradients* across groups, as shown in Figure 1c. By matching gradient variances, our method accounts for within-group variability in the optimization signals induced by individual samples. This encourages more consistent optimization behavior across groups and reduces the risk that a subset of samples disproportionately influences the training dynamics. By integrating these regularizations together (Figure 1d), FairGM harmonizes both the average and the variability of fairness-related optimization signals, leading to more balanced training dynamics and improved fairness–accuracy trade-offs.

These regularizations together encourage the optimization trajectory to evolve similarly for different groups, leading to more stable fairness outcomes and improved fairness–accuracy trade-offs. Considering the trade-offs between task performance and fairness, and our design of three different moments of fairness gradient matching, it could be cumbersome work to tune the model and strike a good balance. Rather than approaching this with traditional methods of linear weighting of task loss and fairness constraints, we reframe the problem through the lens of multi-objective optimization. To effectively balance predictive performance and fairness objectives, we formulate the training problem as a preference-aware multi-objective optimization problem and adopt a Pareto-based optimization scheme EPO (Mahapatra & Rajan, 2020). Given a user-specified preference vector, the preference-aware EPO solver automatically determines the descent direction that leads to the corresponding Pareto-optimal solution, eliminating the need for exhaustive hyperparameter search over regularization weights.

Our proposed framework is compatible with a range of differentiable fairness metrics and gradient-based learning models. We provide the theoretical insights towards the design of each regularization term to elucidate how each of them benefits fairness. Extensive experiments on both synthetic and real-world datasets demonstrate that FairGM consistently achieves favorable fairness–accuracy trade-offs compared to existing fairness-aware learning methods. Our contributions can be summarized as follows:

- We propose a new perspective for fairness-aware learning that considers the alignment of group-conditioned optimization signals rather than focusing solely on metric-level parity.

- We introduce a gradient matching framework that aligns fairness-related gradient statistics across groups under multiple moments, including metric disparity, mean gradients, and gradient variances.

- We formulate fairness-aware learning as a preference-aware multi-objective optimization problem and develop an automated optimization framework to obtain desired fairness–performance trade-offs.

- Extensive experiments on various datasets demonstrate that our method achieves strong and consistent improvements in fairness while maintaining competitive predictive performance.

The structure of the rest of the paper is as follows. We first review the related works in Section 2. Our proposed framework is presented in Section 3 and the theoretical analysis for fairness is provided in Section 4. Experimental results and analysis are outlined in Section 5. Finally, we conclude our paper in Section 6.

## 2 Related Work

### 2.1 Fairness in Machine Learning

Fairness has attracted increasing interest in recent research. Typically, algorithmic group fairness requires the equity of certain fairness metrics over different groups. For instance, No Disparate Treatment (Barocas & Selbst, 2016) requires the same prediction of the example regardless of the group information. Accuracy Parity (Zafar et al., 2017a) requires the same accuracy in predictions for different groups. Equal opportunity demands parity in true positive rates among groups while Equalized Odds simultaneously considers false positive rates (Hardt et al., 2016). While these different fairness notions have been compared by researchers such as in Zafar et al. (2019), there is no consensus regarding which definition is universally the most appropriate. To satisfy the respective notion, various algorithmic solutions (Zafar et al., 2017b; Donini et al., 2018; Agarwal et al., 2018) have been designed to address the risk minimization problem with the fairness constraints. However, these algorithmic solutions are often deemed hard to comprehend (Saha et al.,

2020) or put into practice (Beutel et al., 2019). Another line of work of fairness is to debias in machine learning models. Locatello et al. (2019) disentangle latent factors and segregate the sensitive group factor from the representation. Madras et al. (2018) and Zhang et al. (2018) suggest bias mitigation through adversarial means. Shui et al. (2022) designs a bi-level optimization framework of implicit path alignment which pursues the same predictor for different groups based on the spirit of invariant representation learning (Arjovsky et al., 2019). Maheshwari & Perrot (2023) designed dynamically weighted loss for groups with group-level fairness violations. Li & Liu (2022) and Wang et al. (2024) reweight the samples in groups leveraging the influence function. Although these methods can be readily tailored to different deep-learning models based on gradient descent like ours, they do not exploit the gradient information on the groups, which is a potent representation in neural networks.

## 2.2 Gradient Matching

The efficacy of aligning gradients across varying domains or tasks is well established across multiple learning settings. Lopez-Paz & Ranzato (2017) have demonstrated the utility of gradient alignment in continual learning contexts, by evaluating whether a gradient update could potentially increase the loss of previous tasks. Zhang et al. (2019) devise a strategy that incorporates an angle bisector between the task and novelty gradients, proving effective across a variety of tasks in reinforcement learning. Shi et al. (2022) encourages gradients from different domains to be consistent by maximizing the gradient inner product, thereby enhancing the model's generalization capabilities to unseen domains. Rame et al. (2022) further matches the domain-level gradient variances and mitigates inconsistencies across domains. (Ballas & Diou, 2025) optimize the neural networks with domain-aligned gradients for domain-invariant solutions. In fair classification tasks, Li et al. (2023) minimizes the magnitude difference of the group gradients in the task loss space to promote fairness. Alternatively, Malik & Mopuri (2025) implicitly aligns task gradients in different subpopulations in a meta-learning way to achieve fairness. Castiglione et al. (2022) pursues a small $l_2$ norm of the integrated gradient of the classification model and auxiliary model (which is to predict the sensitive attribute) to ensure that neither of them is biased towards sensitive features. In this work, we instead explore beyond the task loss space, designing gradient matching regularizations of varying degrees of granularity within the fairness metric space to boost fairness.

## 2.3 Multi-objective Optimization

Multi-objective optimization (MOO) provides solutions for optimizing a set of objectives. Comprehensive surveys of this field can be found in the works of Marler & Arora (2004); Deb & Deb (2013) and Wiecek et al. (2016). Of particular relevance to our work, gradient-based MOO methods, as developed by Désidéri (2012); Fliege & Vaz (2016); Peitz & Dellnitz (2018), use multi-objective Karush-Kuhn-Tucker conditions and find a descent direction that decreases all objectives. Furthermore, various kinds of preferences, such as objective weights, goal specification and desirability thresholds, can be incorporated in a MOO. Notably, EPO (Mahapatra & Rajan, 2020) combines gradient descent and carefully controlled ascent to pinpoint an exact preference-specific Pareto optimal solution. Although MOO has been widely applied to improving multi-task learning (Sener & Koltun, 2018; Lin et al., 2019), it remains under-explored on how to model and optimize fairness objectives from the preference-aware MOO perspective.

# 3 Proposed Framework

In this section, we begin with the background of this work. Then, we introduce the design of each fair regularization term. Finally, we present our proposed optimization framework FAIRGM.

## 3.1 Background

Consider a classification task: given a data set $\{(\boldsymbol{x}^i, y^i)\}_{i=1}^n$ of $n$ samples $\boldsymbol{x}$ and their corresponding labels $y$, we aim to learn a model $f_\theta$ which consists of a representation function $\Phi : \mathcal{X} \to \mathcal{H}$ and the predictor $\rho : \mathcal{H} \to \mathcal{Y}$, where $\mathcal{X}, \mathcal{H}, \mathcal{Y}$ denote the input space, hidden space and output space respectively. For a binary classification task, we have $\mathcal{Y} = \{0, 1\}$. Let $s \in \mathcal{S}$ be a sensitive group and $n_s$ be the number of samples

in group $s$. Then, we use $\{(\boldsymbol{x}_s^i, y_s^i)\}_{i=1}^{n_s}$ to represent all the samples in group $s$. For notation simplicity, we illustrate our method in the binary sensitive attribute case in the next sections, i.e., $\mathcal{S} = \{0, 1\}$, although our proposed techniques can be naturally extended to the multiple-group case. Classical methods minimize the task-specific risk $\mathcal{R}$, such as cross-entropy loss, for the best task performance.

Algorithmic fairness usually requires equity of a certain fairness metric $\mathcal{M}$ over groups. While various definitions of fairness have been proposed so far (Chouldechova & Roth, 2018), without loss of generality, we define $\mathcal{M}_s(f) = \frac{1}{n_s} \sum_{i=1}^{n_s} \mathcal{M}(f(\boldsymbol{x}_s^i), y_s^i)$ as the group-level fairness metric of model $f$ over each group $s$. We thus can naturally use the difference between the metrics on the two groups $|\mathcal{M}_0(f) - \mathcal{M}_1(f)|$ to evaluate the fairness level of the model $f$.

## 3.2 Fair Gradient Matching

In this subsection, we will introduce the detailed design of the fair gradient matching regularization terms under the three different moments. We organize our regularizations by the order of gradient statistics they align: the fairness metric value itself (zeroth order, no gradient), the mean gradient (first order), and the gradient variance (second order). We use the term 'moment' loosely to refer to these levels of statistical granularity, following conventions in gradient-based learning literature.

### 3.2.1 Zeroth-moment Fair Optimization

The most intuitive method for fair optimization is to add the difference in the fairness metrics of the two groups as the regularization in the optimization objective. That is,

$$FairReg_0 = |\mathcal{M}_0(f) - \mathcal{M}_1(f)|^p, \tag{1}$$

where $p \in \mathbb{Z}^+$ and we choose $p = 2$ in our implementation. Since it considers no gradient of the model, we refer to it as zeroth-moment fair optimization. This is the most direct optimization targeting the desired fairness metric. Instead of relying on computationally intensive analytical solutions, we optimize the difference with the ubiquitous gradient descent procedure in neural networks, which is more flexible and efficient.

In the case of non-differentiable fairness metrics, such as Equal Opportunity (Hardt et al., 2016) (i.e., the same true positive rate across groups), we can use the reparameterization trick (Jang et al., 2022) or the surrogate functions. We refer the readers to Lohaus et al. (2020) for more discussions of the design and use of surrogate functions for fairness metrics.

### 3.2.2 First-moment Fair Optimization

As substantiated by recent works in continual learning (Lopez-Paz & Ranzato, 2017), reinforcement learning (Zhang et al., 2019), domain generalization (Shi et al., 2022) and multi-task learning (Yu et al., 2020), the endeavor to enhance gradient alignment emerges as an indispensable strategy for improving performance across diverse distributions. Inspired by this, we propose to match the gradients in the fairness metric space to reduce discrimination against groups. Let the individual gradient $\boldsymbol{g}_s^i = \nabla \mathcal{M}(f_\theta(\boldsymbol{x}_s^i, y_s^i))$ be the derivative for the $i$-th sample from group $s$, and $\boldsymbol{G}_s = [\boldsymbol{g}_s^i]_{i=1}^{n_s}$ of size $n_s \times |\theta|$ be the corresponding group-based gradient matrix, where $|\theta|$ is the size of the model parameters $\theta$. We can compute the group-level mean gradient as $\overline{\boldsymbol{g}}_s = \frac{1}{n_s} \sum_{i=1}^{n_s} \boldsymbol{g}_s^i$ for each group $s$. To promote the alignment between the gradients, we propose to minimize the $L_2$ difference in mean gradients, which is formally written as:

$$FairReg_1 = ||\overline{\boldsymbol{g}}_0 - \overline{\boldsymbol{g}}_1||_2^2 \tag{2}$$

Since this term is based on the mean gradients, we refer to it as first-moment fair optimization.

### 3.2.3 Second-moment Fair Optimization

Considering that simply using the mean gradients can hardly capture the individual gradient information in the group, we may look one step further. A notable study by Rame et al. (2022) offers a compelling direction.

It underscores that minimizing the domain gradient variance difference of the task loss emerges as a more potent strategy for generalization across domains. In our setting, we instead delve deeper into the fairness metric space, exploring the group-specific gradient variances of the fairness metric, which is computed as:

$$\boldsymbol{v}_s = \text{Var}\left(\boldsymbol{G}_s\right) = \frac{1}{n_s} \sum_{i=1}^{n_s} \left(\boldsymbol{g}_s^i - \overline{\boldsymbol{g}}_s\right)^2 \tag{3}$$

where element-wise product is used for the square function. To pursue a more fine-grained gradient alignment in the fairness metric space, we aim to make the group-level variances similar to each other and thus formulate our regularization term as

$$FairReg_2 = ||\boldsymbol{v}_0 - \boldsymbol{v}_1||_2^2 \tag{4}$$

Since this term is based on the gradient variances, we refer to it as second-moment fair optimization.

We adopt element-wise gradient variance rather than the full covariance matrix for computational efficiency, as the latter requires $O(|\theta|^2)$ storage and computation per group. This choice follows Rame et al. (2022) and, as shown in Proposition 4.1, is sufficient to enforce second-moment distributional alignment under multiple feature-induced weighting schemes. Exploring structured covariance approximations (e.g., block-diagonal or low-rank) is a promising direction for future work.

**Remark.** For notation simplicity, we present FairGM in the binary sensitive attribute case throughout the paper. The framework extends naturally to multiple groups by replacing the pairwise difference with a sum over all group pairs. Each regularization remains a differentiable scalar objective, so the MOO formulation and EPO solver require no modification. We demonstrate the effectiveness of this extension in Section 5.7.

### 3.2.4 Optimization Scheme for FairGM

As observed in many works (Wang et al., 2021), there is a trade-off between task performance and fairness. Let us denote the common task performance loss in empirical risk minimization (ERM) as $\mathcal{L}_{\text{ERM}} = \frac{1}{n} \sum_{i=1}^{n} \mathcal{R}(f(\boldsymbol{x}^i), y^i)$. To train a fair model, a typical training scheme is the minimization of a weighted sum of $\mathcal{L}_{\text{ERM}}$ and the fairness regularizations presented in the previous subsections. Formally,

$$\min_{\theta} \quad \mathcal{L}_{\text{ERM}} + \lambda_0 FairReg_0 + \lambda_1 FairReg_1 + \lambda_2 FairReg_2 \tag{5}$$

However, it can be tedious work to tune the weights $(\lambda_0, \lambda_1, \lambda_2)^T$ for each term to reach the desired trade-off. To tackle this problem, we instead view the optimization problem (5) from the perspective of multi-objective optimization (MOO) (Miettinen, 1999). MOO considers solving $k$ objectives w.r.t. $\{\mathcal{L}_i\}_{i=1}^k$ losses, i.e.,

$$\min_{\theta} \mathcal{L}(\theta) = (\mathcal{L}_1(\theta), \dots, \mathcal{L}_k(\theta))^T.$$

A solution $\theta$ dominates another $\theta'$ if $\mathcal{L}_i(\theta) \leq \mathcal{L}_i(\theta')$ for all $i$ and $\mathcal{L}(\theta) \neq \mathcal{L}(\theta')$. A solution $\theta^*$ is called *Pareto optimal* if no other $\theta$ dominates $\theta^*$. The set of Pareto optimal solutions is called the Pareto set. We can reformulate problem (5) from the MOO perspective as:

$$\min_{\theta} \quad \mathcal{L}_{\text{MOO}} = (\mathcal{L}_{\text{ERM}}, FairReg_0, FairReg_1, FairReg_2)^T \tag{6}$$

Ideally, the set of Pareto optimal solutions is small, which satisfies $FairReg_0 = FairReg_1 = FairReg_2 = 0$ with a minimal $\mathcal{L}_{\text{ERM}}$ and thus achieves the desired fair solutions. However, achieving these ideal constraints may prove to be excessively challenging due to the presence of noise in features and labels (Ahuja et al., 2021). Therefore, it is natural to relax the constraints by introducing a parameter $\epsilon_i$ such that $FairReg_i \leq \epsilon_i, i \in \{0, 1, 2\}$, where approaching $\epsilon_i$ to zero recovers the ideal fairness. As pointed out in Chen et al. (2023), when $\{\epsilon_i\}_{i=0}^2 > 0$, there can be multiple Pareto optimal solutions while there are very few desired fair solutions. Hence a preference for ERM and fairness objectives is generally necessary. By specifying a preference $\boldsymbol{p} = (p_E, p_0, p_1, p_2)^T$ over the task performance and each fairness regularization, the optimal

solution to Equation (6) satisfies $\boldsymbol{p}_i \mathcal{L}_{\text{MOO}_i} = \boldsymbol{p}_j \mathcal{L}_{\text{MOO}_j}, \forall i, j \in \{E, 0, 1, 2\}$. Intuitively, the ideal preference aligns inversely with each $\epsilon_i$, i.e., $\boldsymbol{p} = \left(\epsilon_E^{-1}, \epsilon_0^{-1}, \epsilon_1^{-1}, \epsilon_2^{-1}\right)^T$. We leave the proof in Appendix A.

We design an optimization framework FAIRGM to solve for this MOO problem, which approaches a Pareto optimal solution that minimizes the fairness regularizations while not sacrificing the task performance too much. FAIRGM consists of two phases, the *warm-up* phase and the *debias* phase. We first warm-up the model by merely minimizing $\mathcal{L}_{\text{ERM}}$ and then debias the model with the Pareto minimal solutions with a given preference $\boldsymbol{p}$ over the task performance and each fairness regularization. We adopt the preference-aware MOO solver EPO (Mahapatra & Rajan, 2020) to find the desired solutions with the given $\boldsymbol{p}$, where the model is updated with the re-weighted descent direction to both improve the satisfaction to the exact preference-aware Pareto optimality, and to avoid increasing all the loss and regularization values. The computation details can be found in Appendix B.

Our proposed FAIRGM reduces the hyperparameter search space from three unconstrained weights ($\lambda_0$, $\lambda_1$, $\lambda_2$) to a single preference vector, which has a more interpretable meaning in terms of desired fairness–performance trade-offs. Notably, as pointed out in Chen et al. (2023), the linear weighting scheme in Equation (5) cannot reach any solutions in the non-convex part of the Pareto front. In contrast, our method finds an adaptive descent direction under gradient conflicts that leads to the desired solution.

To reduce the computational cost, we only update the last layer which is the classifier $\rho$. Extensive research has validated the efficiency and effectiveness of last-layer fine-tuning, as demonstrated by several notable works (Kirichenko et al., 2022; Iwasawa & Matsuo, 2021; Mao et al., 2023). In addition, computing per-sample gradients is efficiently handled using the BackPACK library (Dangel et al., 2020). The EPO solver requires solving a small linear program at each step, for which we use Gurobi with an academic license, and an open-source linear program solver can be substituted with minimal performance impact.

## 4 Fairness with FairGM

In this section, we analyze how each design of fairness regularization promotes model fairness.

### 4.1 Zeroth-moment Regularization for Fairness

In this work, since we assume no distribution shift from the training set to the test set, directly minimizing $|\mathcal{M}_0(f) - \mathcal{M}_1(f)|^2$ provides a necessary condition for fairness: at any solution where $FairReg_0 = 0$, the model achieves exact metric parity across groups. However, metric parity at a single parameter setting does not imply stability with respect to optimization: if the gradient landscapes of the two groups differ around the solution, small parameter changes can quickly reintroduce disparities. Furthermore, $FairReg_0$ constrains only a scalar summary of each group's error distribution, leaving the full distributional structure of prediction errors unconstrained. The first- and second-moment regularizations address both limitations, as we show below.

### 4.2 First-moment Regularization for Fairness

For the first-moment regularization term, following Rame et al. (2022), let us consider the model $f_\theta$ as a linear classifier with parameters of weights $\boldsymbol{w} = \{w_j\}_{j=1}^r$ and bias $b$, $\hat{y}_s^i$ as the predictions after sigmoid, and take the cross-entropy loss as $M_s$ as an example. The mean gradient on group $s$ is computed as:

$$\nabla_{w_j}\mathcal{M}(\boldsymbol{x}_s, \boldsymbol{y}_s) = \frac{1}{n_s}\sum_{i=1}^{n_s}(\hat{y}_s^i - y_s^i)\boldsymbol{x}_s^{i,j}, \quad \nabla_b\mathcal{M}(\boldsymbol{x}_s, \boldsymbol{y}_s) = \frac{1}{n_s}\sum_{i=1}^{n_s}(\hat{y}_s^i - y_s^i), \tag{7}$$

From Equation (7), we can see that the mean gradients in $b$ are the mean prediction residual in the group, and the mean gradients in $w_j$ are the prediction residual weighted by the $j$-th feature. Thus, when we match the mean gradients, we are matching the prediction residual with $r + 1$ different weighting schemes, which depend on the distribution of the features.

### 4.3 Second-moment Regularization for Fairness

Similar to the analysis for $FairReg_1$, consider the model $\theta$ as a linear classifier with parameters $\{w_j\}_{j=1}^r$ and bias $b$, and $M_s$ as the cross-entropy loss. With the gradient of weight $w_j$ and $b$ in Equation (7), we have the gradient variance on group $s$ as:

$$
\begin{aligned}
\boldsymbol{v}_s^{w_j} &= \frac{1}{n_s} \sum_{i=1}^{n_s} ((\hat{y}_s^i - y_s^i)\boldsymbol{x}_s^{i,j})^2 - \Big(\frac{1}{n_s} \sum_{i=1}^{n_s} (\hat{y}_s^i - y_s^i)\boldsymbol{x}_s^{i,j}\Big)^2 \\
\boldsymbol{v}_s^b &= \frac{1}{n_s} \sum_{i=1}^{n_s} (\hat{y}_s^i - y_s^i)^2 - \Big(\frac{1}{n_s} \sum_{i=1}^{n_s} (\hat{y}_s^i - y_s^i)\Big)^2.
\end{aligned}
\tag{8}
$$

The variance terms decompose naturally into two parts. The first parts, $\frac{1}{n_s}\sum_{i=1}^{n_s}((\hat{y}_s^i - y_s^i)\boldsymbol{x}_s^{i,j})^2$, and $\frac{1}{n_s}\sum_{i=1}^{n_s}(\hat{y}_s^i - y_s^i)^2$, correspond to the mean of the squared feature-weighted residuals and the mean squared error respectively. That is, the second moment of the feature-weighted error distribution under each of the $r+1$ weighting schemes. The second parts, $(\frac{1}{n_s}\sum_{i=1}^{n_s}(\hat{y}_s^i - y_s^i)\boldsymbol{x}_s^{i,j})^2$ and $(\frac{1}{n_s}\sum_{i=1}^{n_s}(\hat{y}_s^i - y_s^i))^2$, are the squares of the mean gradients already addressed by $FairReg_1$. As further support, Rame et al. (2022) discussed how matching gradient variances promotes the model's generalization ability to different distributions under more complex cases, and their analysis can be naturally adapted to our fair classification task as another perspective to show how aligning group-specific gradient variances helps reduce disparities in the fairness metrics across groups.

**Proposition 4.1** (Gradient Matching as Distributional Error Matching). *For a linear classifier with cross-entropy loss, let $\mathcal{W} = \{x^j\}_{j=1}^r \cup \{1\}$ denote the set of $r+1$ feature-induced weighting schemes, where $x^j$ denotes the $j$-th feature and $1$ denotes uniform weighting. Then:*

(i) ***First-moment matching:*** *$FairReg_1 = 0$ implies that the first moment of the feature-weighted prediction error distribution is matched across groups under all weighting schemes in $\mathcal{W}$, i.e., $\frac{1}{n_0}\sum_{i=1}^{n_0}(\hat{y}_0^i - y_0^i)w_0^i = \frac{1}{n_1}\sum_{i=1}^{n_1}(\hat{y}_1^i - y_1^i)w_1^i, \forall w \in \mathcal{W}$, where $w_s^i \in \{x_s^{i,j}, 1\}$ is the weight for the $i$-th sample in group $s$.*

(ii) ***Second-moment matching:*** *$FairReg_1 = FairReg_2 = 0$ further implies that the second moment of the feature-weighted prediction error distribution is matched across groups under all weighting schemes in $\mathcal{W}$, i.e., $\frac{1}{n_0}\sum_{i=1}^{n_0}\left((\hat{y}_0^i - y_0^i)w_0^i\right)^2 = \frac{1}{n_1}\sum_{i=1}^{n_1}\left((\hat{y}_1^i - y_1^i)w_1^i\right)^2, \forall w \in \mathcal{W}$.*

(iii) ***Joint matching:*** *$FairReg_0 = FairReg_1 = FairReg_2 = 0$ implies metric parity, gradient direction alignment, and gradient variability alignment simultaneously, enforcing a progressively richer and nested set of distributional constraints on group-level prediction errors.*

**Remark.** We use the population variance (dividing by $n_s$) rather than the sample variance (dividing by $n_s - 1$) in the definition of $FairReg_2$. This ensures that Proposition 4.1(ii) holds without requiring equal group sizes. In practice, the difference is negligible for the sample sizes in our experiments. Proposition 4.1 shows that the three regularizations enforce a nested family of distributional alignment conditions: $FairReg_0$ constrains a scalar summary, $FairReg_1$ constrains the first moment of the full error distribution under $r+1$ schemes, and $FairReg_2$ further constrains the second moment. This hierarchy justifies the use of all three regularizations jointly, as each adds strictly new alignment constraints not captured by the others. While various works have discussed the trade-off between fairness and task performance (Agarwal et al., 2018; Li & Liu, 2022), and the complementary effect of matching gradients under different moments (Shi et al., 2022; Rame et al., 2022), we further study the trade-off between the task loss and all the regularization terms, and demonstrate the benefit of our design in Section 5. Extending Proposition 4.1 to non-linear models and surrogate losses remains an open theoretical question, though our extensive empirical results demonstrate practical effectiveness beyond the linear case.

### 4.4 Preference-aware MOO for Fairness

Considering that the ideal preference of task performance and the fairness of different granularities are usually unknown, we follow Mahapatra & Rajan (2020) and Chen et al. (2023) to analyze the performance of our proposed FAIRGM and establish the following theorem:

**Theorem 4.2.** *Let $\mathcal{F}$ be a finite hypothesis class. Suppose the ERM and FairReg losses are bounded (Assumption A.3 in Appendix A). Let $I = \{E, 0, 1, 2\}$ index all objectives, $m = |I| - 1$, $p_{\max} := \max_{i \in I} p_i$, and $L_{\max} := \max(L_{\mathrm{ERM}}, L_{\mathrm{FairReg}}^0, L_{\mathrm{FairReg}}^1, (2G_{\max})^2)$. For a given $\epsilon > 0$, define*

$$\delta := \min_{f \in \mathcal{F},\ i,j \in I,\ i \neq j} \Big| |p_i L_i(f) - p_j L_j(f)| - \epsilon \Big|. \tag{9}$$

*Assume $\delta > 0$. If the number of training samples satisfies*

$$|D| \geq \frac{32\, L_{\max}^2\, p_{\max}^2}{\delta^2} \log \frac{2(m+1)|\mathcal{F}|}{\gamma}, \tag{10}$$

*then with probability at least $1 - \gamma$, FAIRGM yields an $\epsilon$-approximated solution of the fair model, i.e., $|p_i L_i(\hat{f}^\epsilon) - p_j L_j(\hat{f}^\epsilon)| \leq \epsilon$ for all $i, j \in I$.*

**Remark.** This theorem provides a sample complexity guarantee: with sufficient data, the solution obtained by optimizing the empirical preference-constrained problem satisfies the same fairness constraints at the population level. Proposition 4.1 provides analytical justification for the design of each regularization term in the tractable linear-classifier setting, while Theorem 4.2 provides a complementary statistical guarantee. Note that the convergence of the EPO solver to the Pareto-optimal solution is established separately in Mahapatra & Rajan (2020). We leave the proof of this theorem in Appendix A. In practice, our model can achieve a satisfactory fair solution on most data sets when we assign relatively larger preferences for the first-moment and second-moment regularizations without per-dataset tuning. A sensitivity analysis is provided in Section F.1. When one has access to some running histories, he or she has some empirical knowledge about the task loss and fairness metric values, and thus could obtain an empirical estimate of the fairness metric values w.r.t. ERM loss values at convergence, so that he or she can design a better preference. We believe that obtaining a better estimate of the ideal preference would be a promising future direction based on our work.

## 5 Experiments

To empirically study the fairness and the task performance achieved with FAIRGM, in this section we answer the following three research questions:

- RQ1: How does FAIRGM compare with other baselines in fairness and task performance?

- RQ2: How does our combined gradient matching compare to each moment gradient matching alone?

- RQ3: How does the design of different metric functions affect fairness?

### 5.1 Experimental Setup

We compare FAIRGM with the following invariant and fair machine learning baselines: IRM (Arjovsky et al., 2019), Reduction (Agarwal et al., 2018), Adv (Zhang et al., 2018), Reweight (Li & Liu, 2022), and FairGrad (Maheshwari & Perrot, 2023). We also compare our method with its variants using zeroth-moment gradient matching only $\mathrm{FAIRGM}_0$, first-moment gradient matching only $\mathrm{FAIRGM}_1$, and second-moment gradient matching only $\mathrm{FAIRGM}_2$. The details of the baselines are in Appendix C. For our method, we use the cross-entropy loss as the task risk function, and use three different loss functions as the fairness metric respectively: (1) Cross-entropy loss (CE): the same loss function as the task risk; (2) Squared error loss (SL): minimizing the difference of SL between groups can be viewed as minimizing a surrogate function of accuracy

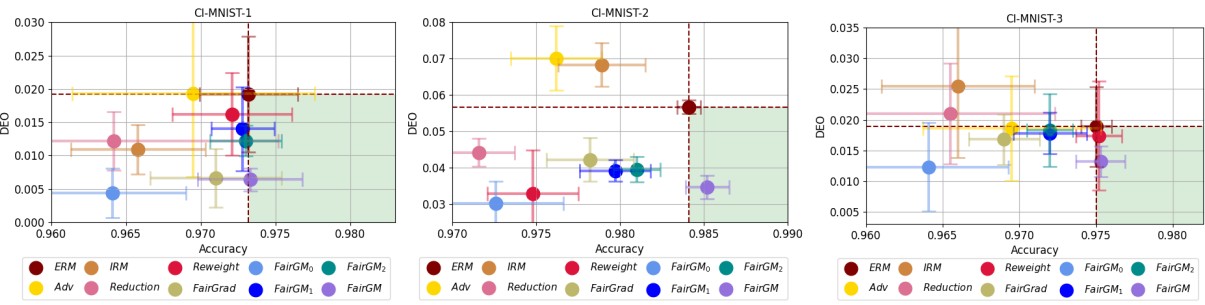

Figure 2: Results on three variants of CI-MNIST. Y-axis shows the gap in true positive rate between the two groups (DEO), while X-axis shows the prediction accuracy. According to the task and fairness performance of the base ERM model, we plot a horizontal and a vertical lines in each figure and divide the space by fairness and task performance results into four regions, where the space in green means a more fair and more accurate model compared to the base ERM model. A point closer to the bottom right indicates better performance in both task performance and fairness.

disparity; (3) True positive rate loss (TPR): we use the surrogate function to compute the non-differentiable TPR (details in Appendix C).

For evaluation, we use Accuracy (Acc) and F1 score as the task performance metric, and use Accuracy Difference (DA) and Equal Opportunity Difference (DEO) as the fairness metric.

## 5.2 Results on Different Types of Bias (RQ1 & RQ2)

We first study how FAIRGM compares with other baseline models in fairness-aware machine learning. We explore this on a synthetic data set. The Correlated and Imbalanced MNIST (CI-MNIST), a variant of the MNIST data set, is firstly proposed by Reddy et al. (2021) as an evaluation benchmark for bias-mitigation approaches, specifically designed to address challenging scenarios and provide control over various data set configurations. This data set offers manual control over different types of bias in the data set, including the number of samples in each group and class, and the group distribution over the class. We keep the train/valid/test splits as the original setup (Reddy et al., 2021) and follow Wang et al. (2024) to examine the models under three types of bias on variants of the data set, including (1) different group sizes, (2) group distribution shifts and (3) different class sizes. The details of the data set are in Appendix D.1. We use a multilayer perceptron with one hidden layer of 64 as the backbone for all the models. For FAIRGM, we use TPR as the fairness metric and assign the relative preference by default as $[1, 5, 20, 20]$. More training details can be found in Appendix E. The mean and the standard error of Accuracy and DEO of different methods of five independent runs are presented in Figure 2.

From the figure we can see that under the different types of bias, the baselines such as IRM and Adv fail to deliver consistent improvement in fairness. The single moment gradient matching methods $\text{FAIRGM}_0$ significantly compromises task performance to enhance fairness. $\text{FAIRGM}_1$ and $\text{FAIRGM}_2$ achieve higher accuracy than $\text{FAIRGM}_0$, but at a reduced level of fairness. To conclude, these baselines struggle to achieve a satisfactory trade-off between task performance and fairness. In contrast, our method consistently strikes a favorable balance between fairness and accuracy with low variance, and notably in some cases, even outperforms all the baselines in both fairness and accuracy.

## 5.3 Comparison with Baselines on Real-world Data Sets (RQ1 & RQ2)

To further study the performance of FAIRGM, we conduct experiments on two commonly-used real-world tabular data sets in fairness, COMPAS (Dieterich et al., 2016) and German (Lichman et al., 2013), and two image data sets, CelebA (Liu et al., 2015) and FairFace (Karkkainen & Joo, 2021). Details of these data sets are in Appendix D. We use a MLP with hidden dimensions of [8, 16] for COMPAS data set and [24, 48] for German data set. For the image data sets, we adopt the PyTorch (Paszke et al., 2017) version of

Table 1: Fair classification in real-world tabular data sets.

| Models | COMPAS | | | | German | | | |
|---|---|---|---|---|---|---|---|---|
| | Acc ↑ | F1 ↑ | DA ↓ | DEO ↓ | Acc ↑ | F1 ↑ | DA ↓ | DEO ↓ |
| ERM | $0.675_{\pm0.011}$ | $0.620_{\pm0.032}$ | $0.027_{\pm0.008}$ | $0.298_{\pm0.048}$ | $0.778_{\pm0.012}$ | $0.592_{\pm0.013}$ | $0.068_{\pm0.012}$ | $0.135_{\pm0.094}$ |
| IRM | $0.667_{\pm0.015}$ | $0.618_{\pm0.026}$ | $0.012_{\pm0.006}$ | $0.268_{\pm0.055}$ | $0.729_{\pm0.015}$ | $0.537_{\pm0.020}$ | $0.043_{\pm0.018}$ | $0.102_{\pm0.084}$ |
| Reduction | $0.669_{\pm0.019}$ | $0.626_{\pm0.040}$ | $0.023_{\pm0.010}$ | $0.206_{\pm0.098}$ | $0.726_{\pm0.025}$ | $0.521_{\pm0.020}$ | $0.051_{\pm0.024}$ | $0.079_{\pm0.060}$ |
| Adv | $0.654_{\pm0.014}$ | $0.617_{\pm0.023}$ | $0.015_{\pm0.010}$ | $0.193_{\pm0.094}$ | $0.758_{\pm0.026}$ | $0.519_{\pm0.021}$ | $0.022_{\pm0.018}$ | $0.094_{\pm0.053}$ |
| Reweight | $0.672_{\pm0.008}$ | $0.643_{\pm0.007}$ | $0.013_{\pm0.004}$ | $0.176_{\pm0.039}$ | $0.747_{\pm0.016}$ | $0.441_{\pm0.028}$ | $0.020_{\pm0.011}$ | $0.056_{\pm0.028}$ |
| FairGrad | $0.671_{\pm0.006}$ | $0.625_{\pm0.016}$ | $0.015_{\pm0.011}$ | $0.181_{\pm0.042}$ | $0.763_{\pm0.021}$ | $0.528_{\pm0.018}$ | $0.027_{\pm0.015}$ | $0.088_{\pm0.046}$ |
| FAIRGM$_0$ | $0.665_{\pm0.017}$ | $0.623_{\pm0.024}$ | $0.018_{\pm0.008}$ | $0.108_{\pm0.063}$ | $0.757_{\pm0.019}$ | $0.534_{\pm0.015}$ | $0.033_{\pm0.018}$ | $0.061_{\pm0.027}$ |
| FAIRGM$_1$ | $0.673_{\pm0.006}$ | $0.627_{\pm0.015}$ | $0.016_{\pm0.009}$ | $0.155_{\pm0.043}$ | $0.766_{\pm0.016}$ | $0.545_{\pm0.014}$ | $0.026_{\pm0.009}$ | $0.034_{\pm0.018}$ |
| FAIRGM$_2$ | $0.670_{\pm0.005}$ | $0.624_{\pm0.018}$ | $0.013_{\pm0.007}$ | $0.134_{\pm0.059}$ | $0.770_{\pm0.016}$ | $0.572_{\pm0.016}$ | $0.030_{\pm0.010}$ | $0.068_{\pm0.022}$ |
| FAIRGM | $0.676_{\pm0.006}$ | $0.641_{\pm0.010}$ | $0.006_{\pm0.004}$ | $0.123_{\pm0.036}$ | $0.771_{\pm0.008}$ | $0.570_{\pm0.010}$ | $0.028_{\pm0.009}$ | $0.042_{\pm0.010}$ |

Table 2: Fair classification in real-world image data sets.

| Models | CelebA | | | | FairFace | | | |
|---|---|---|---|---|---|---|---|---|
| | Acc ↑ | F1 ↑ | DA ↓ | DEO ↓ | Acc ↑ | F1 ↑ | DA ↓ | DEO ↓ |
| ERM | $0.954_{\pm0.007}$ | $0.932_{\pm0.014}$ | $0.045_{\pm0.016}$ | $0.498_{\pm0.041}$ | $0.867_{\pm0.019}$ | $0.843_{\pm0.020}$ | $0.072_{\pm0.022}$ | $0.061_{\pm0.023}$ |
| IRM | $0.949_{\pm0.010}$ | $0.930_{\pm0.008}$ | $0.041_{\pm0.010}$ | $0.351_{\pm0.037}$ | $0.844_{\pm0.024}$ | $0.838_{\pm0.017}$ | $0.062_{\pm0.019}$ | $0.048_{\pm0.014}$ |
| Reduction | $0.947_{\pm0.019}$ | $0.936_{\pm0.021}$ | $0.043_{\pm0.020}$ | $0.346_{\pm0.029}$ | $0.846_{\pm0.026}$ | $0.832_{\pm0.018}$ | $0.051_{\pm0.026}$ | $0.014_{\pm0.012}$ |
| Adv | $0.951_{\pm0.012}$ | $0.937_{\pm0.007}$ | $0.042_{\pm0.014}$ | $0.403_{\pm0.027}$ | $0.866_{\pm0.024}$ | $0.846_{\pm0.028}$ | $0.046_{\pm0.011}$ | $0.013_{\pm0.009}$ |
| FairGrad | $0.951_{\pm0.006}$ | $0.938_{\pm0.012}$ | $0.043_{\pm0.011}$ | $0.342_{\pm0.017}$ | $0.861_{\pm0.021}$ | $0.840_{\pm0.014}$ | $0.042_{\pm0.028}$ | $0.015_{\pm0.010}$ |
| FAIRGM$_0$ | $0.943_{\pm0.012}$ | $0.929_{\pm0.016}$ | $0.042_{\pm0.014}$ | $0.324_{\pm0.032}$ | $0.859_{\pm0.022}$ | $0.831_{\pm0.018}$ | $0.049_{\pm0.015}$ | $0.008_{\pm0.006}$ |
| FAIRGM$_1$ | $0.951_{\pm0.010}$ | $0.935_{\pm0.011}$ | $0.041_{\pm0.017}$ | $0.350_{\pm0.021}$ | $0.864_{\pm0.017}$ | $0.844_{\pm0.019}$ | $0.043_{\pm0.013}$ | $0.012_{\pm0.007}$ |
| FAIRGM$_2$ | $0.953_{\pm0.009}$ | $0.937_{\pm0.009}$ | $0.039_{\pm0.011}$ | $0.338_{\pm0.035}$ | $0.862_{\pm0.018}$ | $0.840_{\pm0.015}$ | $0.040_{\pm0.018}$ | $0.016_{\pm0.008}$ |
| FAIRGM | $0.953_{\pm0.008}$ | $0.939_{\pm0.011}$ | $0.039_{\pm0.011}$ | $0.317_{\pm0.025}$ | $0.866_{\pm0.011}$ | $0.847_{\pm0.014}$ | $0.040_{\pm0.017}$ | $0.009_{\pm0.005}$ |

ResNet-18 (He et al., 2016) as the feature extractor. We set the learning rate at 0.0002 without any learning rate scheduling and set the final layer's hidden dimension to 128. For FAIRGM, we set the fairness metric loss as TPR loss, and assign the relative preference by default as $[1, 5, 20, 20]$. More training details are reported in Appendix E. We conducted five separate runs and aggregated the results in Table 1.

We can observe that many baselines often sacrifice task performance a lot for fairness, such as Adv and Reduction. For the three single moment gradient matching methods, FAIRGM$_0$ usually achieves the best DEO scores since it directly optimizes TPR loss, but at a greater expense of task performance. In contrast, FAIRGM$_1$ and FAIRGM$_2$ usually have better task performance and DA scores than FAIRGM$_0$, indicating the benefit of improving fairness with gradient information. Compared with all the baselines, FAIRGM achieves a favorable trade-off between task performance and fairness across most datasets, with competitive or superior performance on both accuracy and fairness metrics compared to baselines. Notably, it reduces both DA and DEO while maintaining a good or even better accuracy and F1 score than the ERM model.

## 5.4 Weight study (RQ2 & RQ3)

In this subsection, we explore the influence of different fairness metrics on the model, each fairness regularization term's weight on the model, and how our preference-aware solver compares with the manual search of the weight for each regularization. For each fairness metric, we compare our designed preference-aware framework with the weighted training of zeroth-moment FAIRGM$_0$, first-moment FAIRGM$_1$ and second-moment FAIRGM$_2$ fairness regularizations separately. In terms of the regularization weight, we search within the range of $[0.1, 1, 10, 100, 1e3, 1e4]$ to identify the parameter for the best fairness and task performance trade-off on the validation set. The results on COMPAS and FairFace data sets are reported in Table 3.

Table 3: Comparison of regularization terms of different moments with CE, SL and TPR Loss.

| Models | COMPAS | | | | FairFace | | | |
|---|---|---|---|---|---|---|---|---|
| | Acc | F1 | DA | DEO | Acc | F1 | DA | DEO |
| ERM | 0.675 | 0.620 | 0.027 | 0.298 | 0.867 | 0.843 | 0.072 | 0.061 |
| FAIRGM$_0$-CE | 0.668 | 0.648 | 0.011 | 0.224 | 0.858 | 0.834 | 0.032 | 0.029 |
| FAIRGM$_1$-CE | 0.673 | 0.659 | 0.007 | 0.257 | 0.866 | 0.841 | 0.022 | 0.020 |
| FAIRGM$_2$-CE | 0.676 | 0.637 | 0.012 | 0.212 | 0.863 | 0.838 | 0.039 | 0.019 |
| FAIRGM-CE | 0.675 | 0.643 | 0.006 | 0.233 | 0.868 | 0.845 | 0.033 | 0.021 |
| FAIRGM$_0$-SL | 0.665 | 0.638 | 0.005 | 0.218 | 0.861 | 0.840 | 0.025 | 0.024 |
| FAIRGM$_1$-SL | 0.672 | 0.644 | 0.007 | 0.239 | 0.860 | 0.837 | 0.028 | 0.017 |
| FAIRGM$_2$-SL | 0.672 | 0.634 | 0.010 | 0.248 | 0.862 | 0.841 | 0.023 | 0.016 |
| FAIRGM-SL | 0.679 | 0.645 | 0.005 | 0.234 | 0.861 | 0.842 | 0.024 | 0.017 |
| FAIRGM$_0$-TPR | 0.665 | 0.623 | 0.018 | 0.108 | 0.859 | 0.831 | 0.049 | 0.008 |
| FAIRGM$_1$-TPR | 0.673 | 0.627 | 0.016 | 0.155 | 0.864 | 0.844 | 0.043 | 0.012 |
| FAIRGM$_2$-TPR | 0.670 | 0.624 | 0.013 | 0.134 | 0.862 | 0.840 | 0.040 | 0.016 |
| FAIRGM-TPR | 0.676 | 0.641 | 0.006 | 0.123 | 0.866 | 0.847 | 0.040 | 0.009 |

It is evident that when the different granularity levels of fairness regularizations are implemented separately, they each introduce unique trade-offs between task performance and fairness. While FAIRGM$_0$ excels in fostering enhanced fairness metrics, it often does so at the expense of task performance. Conversely, FAIRGM$_1$ and FAIRGM$_2$ usually deliver relatively better task performance but are unable to attain the level of targeted fairness achieved by FAIRGM$_0$. FAIRGM integrates these regularizations and computes the solution in accordance with the desired preference, and achieves the best trade-off. This confirms that the three moments enforce complementary and non-redundant alignment conditions and emphasizes the importance and necessity of combining these regularizations to pursue the desired fair model.

Across fairness metrics, FAIRGM-SL usually achieves the best DA and FAIRGM-TPR ususally achieves the best DEO, consistent with their respective design objectives: SL directly penalizes squared prediction differences which aligns with accuracy disparity, while TPR targets true positive rate equality. Additionally, FAIRGM with SL and TPR often outperforms CE, which suggests that using a fairness metric distinct from the task loss provides a more targeted optimization signal. This highlights a practical guideline: practitioners should select the fairness metric based on the specific fairness criterion they wish to enforce, rather than defaulting to the task loss as a proxy.

### 5.5 Effect of Last-Layer Fine-Tuning (RQ1)

As introduced in Section 3, to reduce the computational cost of our framework, we adopt a last-layer fine-tuning strategy, where the feature extractor remains fixed and only the final classifier layer is updated during fairness optimization. This design significantly reduces the number of trainable parameters and avoids repeatedly backpropagating through the entire network during the multi-objective optimization process.

To validate the effectiveness of this strategy, we compare FAIRGM with its full-training variant (denoted as $FAIRGM_{full}$), where all network parameters are updated. We also include baseline methods for reference. Experiments are conducted on the CI-MNIST benchmark using LeNet (LeCun et al., 2002) as the feature extractor. The results are shown in Figure 3. From the results, we observe that the proposed FAIRGM with last-layer fine-tuning achieves performance comparable to its full-training counterpart across all three CI-MNIST variants. In particular, both variants often fall into the desirable region that improves fairness without sacrificing accuracy relative to ERM. The differences between FAIRGM and $FAIRGM_{full}$ are small, indicating that most of the fairness improvement can be achieved by adjusting the final classifier layer while keeping the learned feature representations fixed. These results suggest that last-layer fine-tuning is an effective and computationally efficient strategy for our framework. It allows FairGM to retain strong fairness–accuracy trade-offs while substantially reducing the training cost compared to full model training.

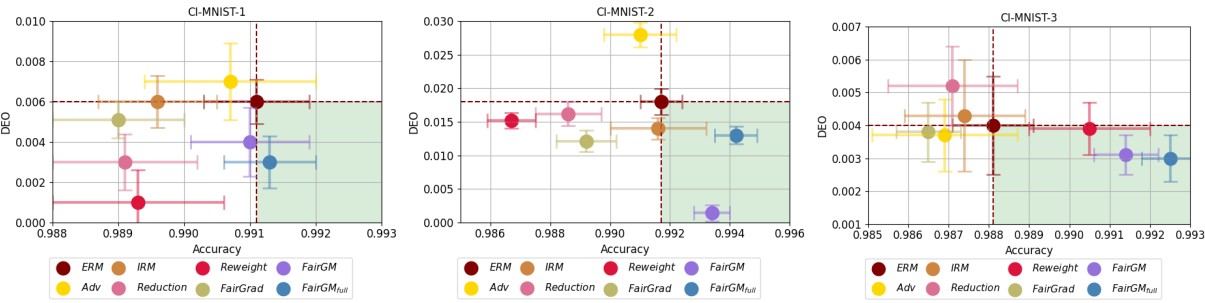

Figure 3: Effect of last-layer fine-tuning.

## 5.6 Effect of Batch Size

In this subsection, we study the influence of batch size on the performance of FairGM. Since our method computes group-wise statistics during training, such as fairness metric differences and gradient statistics across groups, the batch size can affect the stability of these estimates. In particular, very small batch sizes may introduce high variance in the estimation of group-conditioned quantities, potentially leading to unstable optimization. To examine this effect, we evaluate FairGM on the Fair-Face dataset with different batch sizes ranging from 32 to 2048. The results are shown in Figure 4.

From the results, we observe that accuracy and fairness generally improves as the batch size increases, indicating that larger batches provide more stable gradient estimates during training. Notably, performance becomes stably good once the batch size reaches around 256, after which increasing the batch size further yields slight improvements in both accuracy and fairness. Based on these observations, we choose a batch size of 256 as the default setting in our experiments. This value provides a good balance between stable fairness estimation and computational efficiency, while avoiding the increased memory cost associated with very large batch sizes.

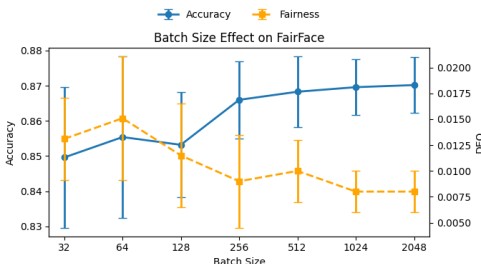

Figure 4: Batch size study.

## 5.7 Extension to Multi-group Scenarios (RQ1)

In the main experiments, we focus on the binary sensitive attribute setting. To validate that FairGM generalizes to the multi-group scenario, we conduct additional experiments with more than two sensitive groups on two data sets Adult (Lichman et al., 2013) and FairFace. For the Adult data set, we construct four disjoint sensitive groups by combining two binary sensitive attributes, race and gender. For FairFace, we extend the binary race attribute to three groups: Black, White, and Latino, resulting in a multi-group fairness setting denoted as FairFace(Multi). We use a MLP with hidden dimensions of $[24, 48]$ for the Adult dataset and adopt the same ResNet-18 backbone as in Section 5.3 for FairFace(Multi). For fairness evaluation, we report DA and DEO as the maximum pairwise difference across all groups. All other training details follow Section 5.3, and we use the last-layer fine-tuning strategy and set the default preference as $[1, 5, 20, 20]$ for FairGM.

The results are presented in Table 4. On the Adult dataset, FairGM achieves the highest accuracy and F1 score among all methods while attaining competitive fairness metrics. On FairFace(Multi), FairGM achieves the best accuracy, F1, and DEO, with DA also among the lowest. Notably, the DEO of FairGM on FairFace(Multi) matches that of the binary-group setting (Table 2), indicating that the method scales to multiple groups without degradation in fairness performance.

Table 4: Results in the multi-group scenario.

| Models | Adult | | | | FairFace(Multi) | | | |
|---|---|---|---|---|---|---|---|---|
| | Acc ↑ | F1 ↑ | DA ↓ | DEO ↓ | Acc ↑ | F1 ↑ | DA ↓ | DEO ↓ |
| ERM | $0.843_{\pm 0.015}$ | $0.698_{\pm 0.029}$ | $0.061_{\pm 0.009}$ | $0.042_{\pm 0.014}$ | $0.862_{\pm 0.021}$ | $0.831_{\pm 0.016}$ | $0.075_{\pm 0.019}$ | $0.069_{\pm 0.018}$ |
| Reduction | $0.808_{\pm 0.021}$ | $0.691_{\pm 0.017}$ | $0.042_{\pm 0.011}$ | $0.033_{\pm 0.012}$ | $0.850_{\pm 0.024}$ | $0.824_{\pm 0.017}$ | $0.048_{\pm 0.021}$ | $0.023_{\pm 0.016}$ |
| Adv | $0.839_{\pm 0.013}$ | $0.696_{\pm 0.014}$ | $0.047_{\pm 0.008}$ | $0.025_{\pm 0.015}$ | $0.854_{\pm 0.016}$ | $0.828_{\pm 0.016}$ | $0.046_{\pm 0.014}$ | $0.017_{\pm 0.009}$ |
| FairGrad | $0.832_{\pm 0.011}$ | $0.695_{\pm 0.012}$ | $0.014_{\pm 0.010}$ | $0.018_{\pm 0.012}$ | $0.858_{\pm 0.014}$ | $0.832_{\pm 0.012}$ | $0.040_{\pm 0.020}$ | $0.019_{\pm 0.012}$ |
| FAIRGM$_0$ | $0.828_{\pm 0.017}$ | $0.693_{\pm 0.018}$ | $0.021_{\pm 0.011}$ | $0.011_{\pm 0.009}$ | $0.851_{\pm 0.019}$ | $0.822_{\pm 0.019}$ | $0.051_{\pm 0.018}$ | $0.014_{\pm 0.009}$ |
| FAIRGM$_1$ | $0.837_{\pm 0.007}$ | $0.697_{\pm 0.010}$ | $0.016_{\pm 0.008}$ | $0.015_{\pm 0.009}$ | $0.859_{\pm 0.017}$ | $0.830_{\pm 0.014}$ | $0.044_{\pm 0.013}$ | $0.016_{\pm 0.010}$ |
| FAIRGM$_2$ | $0.839_{\pm 0.013}$ | $0.702_{\pm 0.011}$ | $0.013_{\pm 0.007}$ | $0.017_{\pm 0.010}$ | $0.857_{\pm 0.015}$ | $0.831_{\pm 0.016}$ | $0.039_{\pm 0.009}$ | $0.012_{\pm 0.006}$ |
| FAIRGM | $0.840_{\pm 0.009}$ | $0.704_{\pm 0.013}$ | $0.014_{\pm 0.009}$ | $0.015_{\pm 0.011}$ | $0.860_{\pm 0.013}$ | $0.833_{\pm 0.010}$ | $0.037_{\pm 0.006}$ | $0.009_{\pm 0.008}$ |

The ablation pattern is consistent with our binary-group findings: FAIRGM$_0$ achieves the lowest DEO by directly targeting the fairness metric, but at the cost of reduced task performance; FAIRGM$_1$ and FAIRGM$_2$ offer better accuracy but less fairness improvement individually; and the full FAIRGM achieves the best overall trade-off by combining all three moments through the preference-aware MOO framework. These results confirm that the pairwise extension mentioned in Section 3 is effective in practice and that FAIRGM maintains its favorable fairness-accuracy trade-offs in the multi-group setting.

**Acknowledgments**

This work is supported by National Science Foundation under Award No. IIS-2416070. The views and conclusions are those of the authors and should not be interpreted as representing the official policies of the funding agencies or the government.

## 6 Conclusion

In this work, we propose FAIRGM, a fairness-aware optimization framework that harmonizes group-conditioned gradient statistics during training. We show theoretically that first- and second-moment gradient matching in the fairness metric space enforces distributional alignment of prediction errors across groups under multiple feature-induced weighting schemes, providing a richer set of fairness constraints than metric-level parity alone. We further formulate training as a preference-aware multi-objective optimization problem, enabling flexible and interpretable trade-offs between predictive performance and fairness. Experiments across tabular and image datasets demonstrate that FairGM generally achieves favorable fairness–accuracy trade-offs compared with existing methods, with consistent behavior across diverse bias settings and dataset scales.

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

# A Proofs

## A.1 Ideal Preference

Let $f_{\text{Fair}}$ represent the desired fair solution, and let $\mathcal{F}$ denote the functional class of $f_{\text{Fair}}$. We consider a group of fair objectives $\boldsymbol{L}_{\text{FairReg}} = \{FairReg_i\}_{i=0}^2$ to be robust if their composite objective $\boldsymbol{L}_{\text{FairReg}}$ satisfies the following condition:

$$\boldsymbol{L}_{\text{FairReg}}(f_{\text{Fair}}) \preceq \boldsymbol{L}_{\text{FairReg}}(f), \forall f \neq f_{\text{Fair}} \in \mathcal{F} \tag{11}$$

When we have a robust fair objective $\boldsymbol{L}_{\text{FairReg}}$, our objective is to solve the following multi-objective optimization (MOO) problem:

$$\min_f \mathcal{L}_{\text{MOO}} = (\mathcal{L}_{\text{ERM}}, \boldsymbol{L}_{\text{FairReg}})^T \tag{12}$$

Here, $\boldsymbol{L}_{\text{FairReg}}$ corresponds to an $\boldsymbol{\epsilon}$-relaxed invariance constraint, defined as $\boldsymbol{L}_{\text{FairReg}}(f_{\text{Fair}}) = \boldsymbol{\epsilon} \preceq \boldsymbol{L}_{\text{FairReg}}(f), \forall f \neq f_{\text{Fair}} \in \mathcal{F}$. We denote the empirical loss of using the underlying invariant features to predict labels as $\epsilon_{\text{ERM}}$. The optimal values of the desired fair solution with respect to Equation (12) can be represented as $(\epsilon_{\text{ERM}}, \boldsymbol{\epsilon})^T = (\mathcal{L}_{\text{ERM}}(f_{\text{Fair}}), \boldsymbol{L}_{\text{FairReg}}(f_{\text{Fair}}))^T$, which correspond to an ideal preference (or fair preference) for the objectives, denoted as $\boldsymbol{p} = (\epsilon_{\text{ERM}}^{-1}, \boldsymbol{\epsilon}^{-1})^T$. The optimal solutions of Equation (12) that satisfy the exact Pareto optimality, i.e., $\boldsymbol{p}_i \mathcal{L}_i = \boldsymbol{p}_j \mathcal{L}_j, \forall \mathcal{L}_i, \mathcal{L}_j \in \boldsymbol{L}_{\text{MOO}}$, are expected to recover $f_{\text{Fair}}$ as stated in Equation (11).

## A.2 Proof of Theorem 4.2

Following Chen et al. (2023), we first restate the informal version of Theorem 4.2 as the following, while the formal description of will be given in Theorem A.4 with more formal definitions.

**Theorem A.1.** *(Informal) For $\gamma \in (0,1)$ and any $\epsilon, \delta > 0$, if $\mathcal{F}$ is a finite hypothesis class, both ERM and FairReg losses are bounded above, let $I$ be the index of all losses, $m = |I| - 1$, $p_{\max} := \max_{i \in I} p_i$ and $L_{\max} := \max_{i \in I} \mathcal{L}_{MOO_i}$, if the number of training samples $|D| \geq \frac{32 L_{\max}^2 p_{\max}^2}{\delta^2} \log \frac{2(m+1)|\mathcal{F}|}{\gamma}$, then with probability at least $1 - \gamma$, FAIRGM yield an $\epsilon$-approximated solution of the fair model.*

Without loss of generality, given a preference $\boldsymbol{p} = (p_E, p_0, ..., p_{m-1})^T = (\frac{1}{\epsilon_E}, \frac{1}{\epsilon_0}, ..., \frac{1}{\epsilon_{m-1}})^T$, the ERM loss $\mathcal{L}_{\text{ERM}}$ and $m$ FairReg losses (in our method $m = 3$) $(FairReg_0, ..., FairReg_{m-1})$, Equation (6) can be reformulated as

$$\begin{aligned} \boldsymbol{f}_{\text{FAIRGM}} &:= \arg\min_{f \in \mathcal{F}} \quad \mathcal{L}_{\text{ERM}}(f) \\ &\text{s.t.} \quad p_E \mathcal{L}_{\text{ERM}}(f) = p_0 FairReg_0(f) = \cdots = p_{m-1} FairReg_{m-1}(f). \end{aligned} \tag{13}$$

We remark that under the ideal preference, the optimal solution of Equation (13), is also the optimal solution to Equation (6) (i.e., the unconstrained version). We will use $\boldsymbol{f}_{\text{FAIRGM}}$ to differentiate from the solution to the unconstrained version. We focus on Equation (13) for the reason that it is more convenient to establish the discussion on the approximated preference, from the perspective of optimization constraints.

Exactly enforcing the above preference constraint is too restrictive *both practically and theoretically*, instead we incorporate the approximation by relaxing the constraint of the loss values w.r.t. the preference. The $\epsilon$-approximated problem of Equation (13) is as the following

$$\begin{aligned} \boldsymbol{f}_{\text{FAIRGM}}^\epsilon &:= \arg\min_{f \in \mathcal{F}} \quad \mathcal{L}_{\text{ERM}}(f) \\ &\text{s.t.} \quad \forall i, j \in I, i \neq j, |p_i \mathcal{L}_i(f) - p_j \mathcal{L}_j(f)| \leq \epsilon, \end{aligned} \tag{14}$$

where $I := \{E, 0, ..., m-1\}$ is the index set of overall losses. For notation simplicity, we slightly abuse the notations and let $\mathcal{L}_E = \mathcal{L}_{\text{ERM}}, \mathcal{L}_0 = FairReg_0, ..., \mathcal{L}_{m-1} = FairReg_{m-1}$. We denote the relaxed constraint set in Equation (14) as $\boldsymbol{P}_{\text{FAIRGM}}^\epsilon := \{f \mid \forall i, j \in I, i \neq j, |p_i \mathcal{L}_i(f) - p_j \mathcal{L}_j(f)| \leq \epsilon\}$. Clearly, it holds that the solution sets satisfy $\boldsymbol{f}_{\text{FAIRGM}}^0 = \boldsymbol{f}_{\text{FAIRGM}}$.

Then we define the empirical version of the $\epsilon$-approximated problem Equation (14) with preference vector $\boldsymbol{p}$ as follows.

$$\hat{\boldsymbol{f}}_{\text{FairGM}}^{\epsilon} := \underset{f \in \mathcal{F}}{\arg\min} \quad \widehat{\mathcal{L}}_{\text{ERM}}(f)$$

$$\text{s.t.} \quad \forall i,j \in I, i \neq j, \ |p_i \widehat{\mathcal{L}}_i(f) - p_j \widehat{\mathcal{L}}_j(f)| \leq \epsilon. \tag{15}$$

Similarly, we denote the above constraint set as $\widehat{\boldsymbol{P}}_{\text{FairGM}}^{\epsilon} := \{f \mid \forall i,j \in I, i \neq j, \ |p_i \widehat{\mathcal{L}}_i(f) - p_j \widehat{\mathcal{L}}_j(f)| \leq \epsilon\}$.

Assume a finite hypothesis class $\mathcal{F}$ and define

$$\delta = \min_{f \in \mathcal{F}, \forall i,j \in I, i \neq j} \big| |p_i \mathcal{L}_i(f) - p_j \mathcal{L}_j(f)| - \epsilon \big|.$$

First, we recall the definition of $\nu$-representative sample (Shalev-Shwartz & Ben-David, 2014).

**Definition A.2.** A training set $S$ is called $\nu$-representative (w.r.t. domain $\mathcal{X}$, hypothesis $\mathcal{F}$, loss $\ell$ and distribution $\mathcal{D}$) if

$$\forall f \in \mathcal{F}, |\widehat{\mathcal{L}}(f) - \mathcal{L}(f)| \leq \nu,$$

where $\mathcal{L}(f) := \mathbb{E}_{(X,Y) \sim \mathcal{D}}[\ell(f(X), Y)]$ and $\widehat{\mathcal{L}}(f) := \frac{1}{|S|} \sum_{(X_i, Y_i) \in S} \ell(f(X_i), Y_i)$.

Equipped with this definition, we can now characterize the condition under which the constraint sets in Equation (14) and Equation (15) contain exact the same predictors.

**Lemma 1.** *For any $\epsilon > 0$, assuming $\delta > 0$ and denoting $p_{\max} := \max_{i \in I} p_i$, if the training set $\mathcal{D}$ is $\frac{\delta}{4p_{\max}}$-representative w.r.t. domain $\mathcal{X}$, hypothesis $\mathcal{F}$, distribution $\mathcal{D}$ and all the ERM and FairReg losses $\{\mathcal{L}_{ERM}, FairReg_0, ..., FairReg_{m-1}\}$, then $\boldsymbol{P}_{FairGM}^{\epsilon} = \widehat{\boldsymbol{P}}_{FairGM}^{\epsilon}$.*

**Remark.** The margin parameter $\delta$ is determined by $(\mathcal{F}, \{\mathcal{L}_i\}, \mathbf{p}, \epsilon)$ and is not user-specified. It measures how well-separated every hypothesis in $\mathcal{F}$ is from the $\epsilon$-boundary of the preference constraint. For a finite $\mathcal{F}$, the condition $\delta > 0$ is mild: it fails only for the finitely many values of $\epsilon$ at which some hypothesis lies exactly on the boundary.

*Proof.* We first show that $\boldsymbol{P}_{\text{FairGM}}^{\epsilon} \subseteq \widehat{\boldsymbol{P}}_{\text{FairGM}}^{\epsilon}$. By the definition of $\delta$, for all $f \in \mathcal{F}$, and $\forall i,j \in I, i \neq j$ we have

$$|p_i \mathcal{L}_i(f) - p_j \mathcal{L}_j(f)| \leq \epsilon - \delta \ \text{ or } \ |p_i \mathcal{L}_i(f) - p_j \mathcal{L}_j(f)| \geq \epsilon + \delta. \tag{16}$$

Using this property, for any $f \in \boldsymbol{P}_{\text{FairGM}}^{\epsilon}$, we can conclude that $\forall i,j \in I, i \neq j$,

$$|p_i \mathcal{L}_i(f) - p_j \mathcal{L}_j(f)| \leq \epsilon \Rightarrow |p_i \mathcal{L}_i(f) - p_j \mathcal{L}_j(f)| \leq \epsilon - \delta.$$

This inequality further implies that

$$|p_i \mathcal{L}_i(f) - p_i \widehat{\mathcal{L}}_i(f) + p_j \widehat{\mathcal{L}}_j(f) - p_j \mathcal{L}_j(f) + p_i \widehat{\mathcal{L}}_i(f) - p_j \widehat{\mathcal{L}}_j(f)| \leq \epsilon - \delta$$

$$\Rightarrow \big| |p_i \widehat{\mathcal{L}}_i(f) - p_j \widehat{\mathcal{L}}_j(f)| - |p_i \mathcal{L}_i(f) - p_i \widehat{\mathcal{L}}_i(f) + p_j \widehat{\mathcal{L}}_j(f) - p_j \mathcal{L}_j(f)| \big| \leq \epsilon - \delta$$

$$\Rightarrow |p_i \widehat{\mathcal{L}}_i(f) - p_j \widehat{\mathcal{L}}_j(f)| \leq \epsilon - \delta + |p_i \mathcal{L}_i(f) - p_i \widehat{\mathcal{L}}_i(f) + p_j \widehat{\mathcal{L}}_j(f) - p_j \mathcal{L}_j(f)|$$

$$\Rightarrow |p_i \widehat{\mathcal{L}}_i(f) - p_j \widehat{\mathcal{L}}_j(f)| \leq \epsilon - \delta + p_i |\mathcal{L}_i(f) - \widehat{\mathcal{L}}_i(f)| + p_j |\widehat{\mathcal{L}}_j(f) - \mathcal{L}_j(f)|,$$

which is based on the triangle inequality of the absolute value function.

From the definition of $\frac{\delta}{4p_{\max}}$-representative, we have $|\mathcal{L}_i(f) - \widehat{\mathcal{L}}_i(f)| \leq \frac{\delta}{4p_{\max}}, \forall i \in I$. Substituting this in the above inequality, we obtain

$$|p_i \widehat{\mathcal{L}}_i(f) - p_j \widehat{\mathcal{L}}_j(f)| \leq \epsilon - \delta + \frac{p_i \delta}{4p_{\max}} + \frac{p_j \delta}{4p_{\max}}$$

$$\leq \epsilon - \frac{\delta}{2},$$

which implied that $f \in \widehat{\boldsymbol{P}}_{\text{FAIRGM}}^{\epsilon}$.

Then, we prove that $\widehat{\boldsymbol{P}}_{\text{FAIRGM}}^{\epsilon} \subseteq \boldsymbol{P}_{\text{FAIRGM}}^{\epsilon}$.

For any $f \in \widehat{\boldsymbol{P}}_{\text{FAIRGM}}^{\epsilon}$, it holds that $\forall i, j \in I, i \neq j$,

$$|p_i \widehat{\mathcal{L}}_i(f) - p_j \widehat{\mathcal{L}}_j(f)| \leq \epsilon$$
$$\Rightarrow |p_i \widehat{\mathcal{L}}_i(f) - p_i \mathcal{L}_i(f) + p_j \mathcal{L}_j(f) - p_j \widehat{\mathcal{L}}_j(f) + p_i \mathcal{L}_i(f) - p_j \mathcal{L}_j(f)| \leq \epsilon$$
$$\Rightarrow \left| |p_i \mathcal{L}_i(f) - p_j \mathcal{L}_j(f)| - |p_i \widehat{\mathcal{L}}_i(f) - p_i \mathcal{L}_i(f) + p_j \mathcal{L}_j(f) - p_j \widehat{\mathcal{L}}_j(f)| \right| \leq \epsilon$$
$$\Rightarrow |p_i \mathcal{L}_i(f) - p_j \mathcal{L}_j(f)| \leq \epsilon + |p_i \widehat{\mathcal{L}}_i(f) - p_i \mathcal{L}_i(f) + p_j \mathcal{L}_j(f) - p_j \widehat{\mathcal{L}}_j(f)|$$
$$\Rightarrow |p_i \mathcal{L}_i(f) - p_j \mathcal{L}_j(f)| \leq \epsilon + p_i |\widehat{\mathcal{L}}_i(f) - \mathcal{L}_i(f)| + p_j |\mathcal{L}_j(f) - \widehat{\mathcal{L}}_j(f)|$$
$$\Rightarrow |p_i \mathcal{L}_i(f) - p_j \mathcal{L}_j(f)| \leq \epsilon + \frac{p_i \delta}{4 p_{\max}} + \frac{p_j \delta}{4 p_{\max}}$$
$$\Rightarrow |p_i \mathcal{L}_i(f) - p_j \mathcal{L}_j(f)| \leq \epsilon + \frac{\delta}{2},$$

which is again based on the triangle inequality of the absolute value function and the definition of $\frac{\delta}{4 p_{\max}}$-representative. Together with Equation (16), we conclude that $|p_i \mathcal{L}_i(f) - p_j \mathcal{L}_j(f)| \leq \epsilon - \delta \Rightarrow f \in \boldsymbol{P}_{\text{FAIRGM}}^{\epsilon}$, which implies $\widehat{\boldsymbol{P}}_{\text{FAIRGM}}^{\epsilon} \subseteq \boldsymbol{P}_{\text{FAIRGM}}^{\epsilon}$.

Based on the above discussion, we have proven that $\boldsymbol{P}_{\text{FAIRGM}}^{\epsilon} = \widehat{\boldsymbol{P}}_{\text{FAIRGM}}^{\epsilon}$. $\qquad\square$

**Assumption A.3** (Bounded Losses). For all $f \in \mathcal{F}$, $X \in \mathcal{X}$, $Y \in \mathcal{Y}$:

(i) The ERM loss is bounded: $|\ell(f(X), Y)| \leq L_{\text{ERM}} < \infty$.

(ii) For $i \in \{0, 1\}$, the fairness regularization can be written as the expectation of a bounded per-sample loss: $\text{FairReg}_i(f) = \mathbb{E}_{(X,Y) \sim D}[\ell_{\text{FairReg}}^i(f(X), Y)]$ with $|\ell_{\text{FairReg}}^i(f(X), Y)| \leq L_{\text{FairReg}}^i < \infty$.

(iii) For $\text{FairReg}_2$, the per-sample gradients of the fairness metric are bounded: $\|g_s^i\| \leq G_{\max} < \infty$ for all $f \in \mathcal{F}$, all groups $s$, and all samples $i$.

**Remark.** Condition (iii) implies that the U-statistic kernel $h(g^i, g^j) = (g^i - g^j)^2$ underlying the sample variance is bounded by $|h| \leq (2 G_{\max})^2$. This is sufficient for the concentration argument in the proof of Theorem A.4: for $\text{FairReg}_0$ and $\text{FairReg}_1$, we apply standard Hoeffding's inequality; for $\text{FairReg}_2$, we apply Hoeffding's inequality for U-statistics (Hoeffding, 1963; Serfling, 2009) with kernel bound $(2 G_{\max})^2$. Both yield exponential concentration at rate $\exp(-Cn)$, so the proof proceeds identically with $L_{\max} := \max(L_{\text{ERM}}, L_{\text{FairReg}}^0, L_{\text{FairReg}}^1, (2 G_{\max})^2)$. The conditions (i) and (ii) are natural and generally hold for many regularization terms. The bounded-gradient condition (iii) is standard and holds whenever the inputs, the model outputs, and the fairness metric are bounded.

**Theorem A.4.** *For any $\epsilon > 0, \gamma \in (0, 1)$, if Assumption A.3 holds and $\delta > 0$, denoting $I$ as the index of all losses, $m = |I| - 1$, $p_{\max} := \max_{i \in I} p_i$ and $L_{\max} := \max_{i \in I} L_i$, if the number of training samples $|\mathcal{D}| \geq \frac{32 L_{\max}^2 p_{\max}^2}{\delta^2} \log \frac{2(m+1)|\mathcal{F}|}{\gamma}$, then with probability at least $1 - \gamma$, we have for any $f_{\text{FAIRGM}}^{\epsilon} \in \boldsymbol{f}_{\text{FAIRGM}}^{\epsilon}$ and $\hat{f}_{\text{FAIRGM}}^{\epsilon} \in \hat{\boldsymbol{f}}_{\text{FAIRGM}}^{\epsilon}$, $\mathcal{L}_{\text{ERM}}(f_{\text{FAIRGM}}^{\epsilon}) \leq \mathcal{L}_{\text{ERM}}(\hat{f}_{\text{FAIRGM}}^{\epsilon}) \leq \mathcal{L}_{\text{ERM}}(f_{\text{FAIRGM}}^{\epsilon}) + \frac{\delta}{2 p_{\max}}$.*

*Proof.* We proceed by first assuming that the training set $D$ is $\frac{\delta}{4 p_{\max}}$-representative w.r.t. domain $\mathcal{X}$, hypothesis $\mathcal{F}$, distribution $\mathcal{D}$ and all the ERM and FairReg losses $\{\mathcal{L}_{\text{ERM}}, FairReg_0, ..., FairReg_{m-1}\}$, and then we establish the sample complexity required for this condition. From Lemma 1, we know that given this condition and the assumptions in the theorem, $\boldsymbol{P}_{\text{FAIRGM}}^{\epsilon} = \widehat{\boldsymbol{P}}_{\text{FAIRGM}}^{\epsilon}$. Then, since the training set $\mathcal{D}$ is $\frac{\delta}{4 p_{\max}}$-representative w.r.t. the ERM loss $\mathcal{L}_{\text{ERM}}$, we have for any $f_{\text{FAIRGM}}^{\epsilon} \in \boldsymbol{f}_{\text{FAIRGM}}^{\epsilon}$ and $\hat{f}_{\text{FAIRGM}}^{\epsilon} \in \hat{\boldsymbol{f}}_{\text{FAIRGM}}^{\epsilon}$,

$$\left| \mathcal{L}_{\text{ERM}}(f_{\text{FAIRGM}}^{\epsilon}) - \widehat{\mathcal{L}}_{\text{ERM}}(f_{\text{FAIRGM}}^{\epsilon}) \right| \leq \frac{\delta}{4 p_{\max}},$$

$$\left| \mathcal{L}_{\text{ERM}}(\hat{f}_{\text{FAIRGM}}^{\epsilon}) - \widehat{\mathcal{L}}_{\text{ERM}}(\hat{f}_{\text{FAIRGM}}^{\epsilon}) \right| \leq \frac{\delta}{4 p_{\max}}.$$

Moreover, based on the optimality of problem equation 15, we can conclude that

$$\mathcal{L}_{\text{ERM}}(\hat{f}_{\text{FAIRGM}}^{\epsilon}) - \frac{\delta}{4p_{\max}} \le \widehat{\mathcal{L}}_{\text{ERM}}(\hat{f}_{\text{FAIRGM}}^{\epsilon}) \le \widehat{\mathcal{L}}_{\text{ERM}}(f_{\text{FAIRGM}}^{\epsilon}) \le \mathcal{L}_{\text{ERM}}(f_{\text{FAIRGM}}^{\epsilon}) + \frac{\delta}{4p_{\max}}$$

$$\Rightarrow \mathcal{L}_{\text{ERM}}(\hat{f}_{\text{FAIRGM}}^{\epsilon}) \le \mathcal{L}_{\text{ERM}}(f_{\text{FAIRGM}}^{\epsilon}) + \frac{\delta}{2p_{\max}}.$$

Then, using the optimality of problem Equation (14), it holds that

$$\mathcal{L}_{\text{ERM}}(f_{\text{FAIRGM}}^{\epsilon}) \le \mathcal{L}_{\text{ERM}}(\hat{f}_{\text{FAIRGM}}^{\epsilon}) \le \mathcal{L}_{\text{ERM}}(f_{\text{FAIRGM}}^{\epsilon}) + \frac{\delta}{2p_{\max}}.$$

It remains to analyze the sample complexity of ensuring that the training set $\mathcal{D}$ is $\frac{\delta}{4p_{\max}}$-representative w.r.t. $\mathcal{X}$, $\mathcal{F}$, $\mathcal{D}$ and all the ERM and FairReg losses $\{\mathcal{L}_{\text{ERM}}, FairReg_0, ..., FairReg_{m-1}\}$.

For notation simplicity, we slightly abuse the notations and let $L_E = L_{\text{ERM}}, L_0 = L_{\text{FairReg}}^0, ..., L_{m-1} = L_{\text{FairReg}}^{m-1}$. For any $i \in I = \{E, 0, ..., m-1\}$, based on Assumption A.3, we can write $\mathcal{L}_i(f) = \mathbb{E}_{(X,Y)\sim\mathcal{D}}[\ell_i(f(X), Y)]$ and $\widehat{\mathcal{L}}_i(f) = \frac{1}{|D|}\sum_{(X_j,Y_j)\in D}\ell_i(f(X_j), Y_j)$ with $|\ell_i(f(X), Y)| \le L_i \le L_{\max}, \forall f, X, Y$. Using Hoeffding's inequality, we can conclude that for any $f \in \mathcal{F}$,

$$\Pr\left[|\widehat{\mathcal{L}}_i(f) - \mathcal{L}_i(f)| \ge \frac{\delta}{4p_{\max}}\right] \le 2\exp\left(\frac{-|D|\delta^2}{32L_{\max}^2 p_{\max}^2}\right).$$

**Concentration for FairReg$_2$.** For $i \in \{E, 0, 1\}$, the loss $L_i(f)$ is a per-sample average with bounded summands, and the concentration bound follows from standard Hoeffding's inequality as above. For $i = 2$ (corresponding to FairReg$_2$), the empirical gradient variance is a U-statistic of order 2:

$$v_s = \frac{1}{n_s(n_s - 1)}\sum_{i<j}(g_s^i - g_s^j)^2. \tag{17}$$

By Assumption A.3(iii), the kernel $h(g^i, g^j) = (g^i - g^j)^2$ satisfies $|h| \le (2G_{\max})^2$. with $t = \frac{\delta}{4p_{\max}}$, we obtain

$$\Pr\left(|v_s - \mathbb{E}[v_s]| \ge \frac{\delta}{4p_{\max}}\right) \le 2\exp\left(-\frac{\lfloor n_s/2 \rfloor \delta^2}{32(2G_{\max})^4 p_{\max}^2}\right), \tag{18}$$

which provides the same exponential concentration as in the per-sample average case. The remainder of the proof proceeds identically, with $L_{\max}$ redefined as $\max(L_{\text{ERM}}, L_{\text{FairReg}}^0, L_{\text{FairReg}}^1, (2G_{\max})^2)$. Thus,

$$\Pr\left[\exists i \in I, \exists f \in \mathcal{F}, |\widehat{\mathcal{L}}_i(f) - \mathcal{L}_i(f)| \ge \frac{\delta}{4p_{\max}}\right]$$

$$\le \sum_{i\in I}\Pr\left[\exists f \in \mathcal{F}, |\widehat{\mathcal{L}}_i(f) - \mathcal{L}_i(f)| \ge \frac{\delta}{4p_{\max}}\right] \le \gamma.$$

Finally, we can conclude that with probability at least $1 - \gamma$, $\forall i \in I, \forall f \in \mathcal{F}$,

$$\left|\widehat{\mathcal{L}}_i(f) - \mathcal{L}_i(f)\right| \le \frac{\delta}{4p_{\max}},$$

which completes the proof. □

# B  EPO details

In this section, we introduce the optimization algorithm EPO we use as the preference-aware MOO solver. As demonstrated in (Mahapatra & Rajan, 2020), to avoid divergence from the Pareto front, at each step, the optimization direction not only needs to minimize $\mathcal{L}_{\text{MOO}}(f)^T\boldsymbol{p}$, but also needs to avoid ascending all

the loss values. More formally, let $\boldsymbol{G}$ denote the gradient signals produced by $\mathcal{L}_{\text{MOO}}$, at step $t$ it solves the following LP for the objective weights $\beta^*$,

$$\beta^* = \arg\max_{\beta \in \mathcal{S}^4} (\boldsymbol{G}\beta)^T \boldsymbol{g}_p,$$

$$\text{s.t. } (\boldsymbol{G}\beta)^T \boldsymbol{G}_j \geq \boldsymbol{g}_p^T \boldsymbol{G}_j, \forall j \in \bar{J} - J^*,$$

$$(\boldsymbol{G}\beta)^T \boldsymbol{G}_j \geq 0, \forall j \in J^*,$$

where $\mathcal{S}^4 = \left\{ \beta \in \mathbb{R}_+^4 \mid \sum_{i=1}^4 \beta_i = 1 \right\}, \boldsymbol{g}_p$ is the adjustment direction that leads to the preferred Pareto optimal solution by $\boldsymbol{p}, J = \left\{ j \mid G_j^T \boldsymbol{g}_p > 0 \right\}$ are the indices of objectives which do not conflict with $\boldsymbol{g}_p$, $\bar{J} = \left\{ j \mid G_j^T \boldsymbol{g}_p \leq 0 \right\}$ are those have conflicts with $\boldsymbol{g}_p, J^* = \left\{ j \mid \boldsymbol{p}_j \mathcal{L}_{\text{MOO}_j} = \max_{j'} \left( \boldsymbol{p}_{j'} \mathcal{L}_{\text{MOO}_{j'}} \right) \right\}$ is the index of the objective which diverges from the preference most.

(Mahapatra & Rajan, 2020) show that using the following $\boldsymbol{g}_p$ could provably lead the solution to converge to the desired preferred Pareto optimal solution, which is defined as follows

$$\boldsymbol{g}_p = \boldsymbol{p} \odot \left( \log(4\hat{\mathcal{L}}_{\text{MOO}}) - \mu(\mathcal{L}_{\text{MOO}}) \right)$$

where $\odot$ is the element-wise product operator, $\hat{\mathcal{L}}_{\text{MOO}}$ is the normalized loss w.r.t. preference $\boldsymbol{p}$, $\mu(\mathcal{L}_{\text{MOO}})$ is the quantitative divergence of the current solution from the preferred direction:

$$\hat{\mathcal{L}}_{\text{MOO}_i} = \boldsymbol{p}_i \mathcal{L}_{\text{MOO}_i} / \sum \boldsymbol{p}_j \mathcal{L}_{\text{MOO}_j}$$

$$\mu(\mathcal{L}_{\text{MOO}}) = \sum_{i=1}^4 \hat{\mathcal{L}}_{\text{MOO}_i} \log \left( 3\hat{\mathcal{L}}_{\text{MOO}_i} \right) = \text{KL}(\hat{\mathcal{L}}_{\text{MOO}} \mid \frac{1}{3})$$

The convergence analysis can be found in the original paper (Mahapatra & Rajan, 2020).

## C  Experimental Setup

We compare our method with the following baselines in invariant representation learning and fair machine learning:

- IRM (Arjovsky et al., 2019): A learning paradigm that estimates invariant causal predictors from multiple training environments to improve generalization to different distributions.

- Reduction (Agarwal et al., 2018): A reduction approach that yields a randomized classifier with the lowest error subject to the desired fairness.

- Adv (Zhang et al., 2018): An approach that jointly trains a predictor and an adversary to maximize task accuracy while minimizing the ability to infer protected attributes from predictions, thereby enforcing fairness constraints through adversarial learning.

- Reweight: (Li & Liu, 2022): A sample reweighting scheme which computes the weights for training samples to achieve fairness using the influence function.

- FairGrad (Maheshwari & Perrot, 2023): An approach that enforces group fairness by dynamically reweighting training examples based on their groups' advantage or disadvantage during gradient descent, guiding the model toward fairer outcomes with minimal changes to standard training.

We compare with these baselines using the following metrics. For task performance, we use Accuracy (Acc) and F1 score. For fairness level, we use accuracy difference (DA) and true positive rate difference (DEO).

For the TPR loss, we use a sigmoid-based surrogate to approximate the indicator function in the true positive rate. For group $s$:

$$\text{TPR}_s^{\text{surr}}(f) = \frac{\sum_{i:y_s^i=1} \sigma(\tau \cdot f_\theta(x_s^i))}{\sum_{i:y_s^i=1} 1}$$

where $\sigma(\cdot)$ is the sigmoid function and $\tau > 0$ is a temperature parameter. As $\tau \to \infty$, the surrogate converges to the true TPR. We set $\tau = 2$ in the experiments.

## D  Datasets

### D.1  Synthetic Dataset

The Correlated and Imbalanced MNIST (CI-MNIST), a variant of the MNIST dataset, is firstly proposed by Reddy et al. (2021) as an evaluation benchmark for bias-mitigation approaches, specifically designed to address challenging scenarios and provide control over various dataset configurations. In this dataset, each image $x$ is assigned a label $y \in \{-1, 1\}$ indicating whether it represents an odd or even number, while the sensitive attribute $s \in \{0, 1\}$ corresponds to the background color, either blue or red. The original dataset assumes that there is a clean and balanced set for test. In this work, we make the distribution of the train set and test set to be consistent. This dataset offers manual control over different types of bias in the dataset, including the number of samples in each group and class, and the group distribution over the class. We keep the train/valid/test splits as the original setup (Reddy et al., 2021) and follow Wang et al. (2024) to examine the models under three types of bias on variants of the data set. The data statistics of them are presented in Table 5.

Table 5: CI-MNIST Data Statistics.

| Biases | Odd (y=-1) | | Even (y=1) | |
|---|---|---|---|---|
| | Blue (s=0) | Red (s=1) | Blue (s=0) | Red (s=1) |
| Different Group Size | 30245 | 5337 | 29257 | 5161 |
| Group Distribution Shift | 30245 | 5337 | 5163 | 29255 |
| Different Class Size | 17791 | 17791 | 4303 | 4301 |

### D.2  Real-world Datasets

- COMPAS (Dieterich et al., 2016). The task in COMPAS is to predict recidivism from someone's criminal history, jail and prison time, demographics, and COMPAS risk scores, with race as the protected sensitive attribute restricted to black (s=0) and white defendants (s=1).

- German (Lichman et al., 2013). The task is to classify people as having good or bad credit risks by features related to the economic situation, with gender as the sensitive attribute restricted to female (s=0) and male (s=1).

- FairFace (Karkkainen & Joo, 2021). The face image dataset is balanced on race, gender and age. In our work, we take the gender prediction as the task and denote the $y = 1$ as *Male* and $y = -1$ as *Female*. And the sensitive attribute $s$ is set to be the race, where $s = 0$ denotes *Black* and $s = 1$ denotes the *White*. We keep the train/valid/test splits as the original setup.

- CelebA (Liu et al., 2015). The task is to predict the Blond Hair attribute, and the senstive attribute is set to be the Male attribute: being female spurious correlates with having blond hair. The minority groups are (blond, male) and the majority groups are (blond, female). We use the standard train/valid/test splits following Sagawa et al. (2020).

For the tabular data sets, we use 20% of the data as a test set and the remaining 80% as a train set. We further divide the train set into two and keep 25% of the training examples as a validation set. The statistics of the datasets are shown in Table 6.

Table 6: Real-world Data Statistics.

| Data sets | Negative | | Positive | |
|---|---|---|---|---|
| | s=0 | s=1 | s=0 | s=1 |
| German | 201 | 499 | 109 | 191 |
| COMPAS | 1514 | 1281 | 1661 | 822 |
| FairFace | 6894 | 6895 | 8789 | 9823 |
| CelebA | 89931 | 28234 | 82685 | 1749 |

## E    Training details

On the three variants of CI-MNIST, we use a MLP with one hidden layer with a dimension as 64. We set
the learning rate as 0.001, warm-up steps as 4000 and the total step as 10000. For German data set, we
use a MLP with two hidden layers of dimensions [24, 48], and we set the learning rate as 0.001, warm-up
step as 100 and and the total step as 400. For COMPAS data set, we use a MLP with two hidden layers
of dimensions [8, 16], and we set the learning rate as 0.001, warm-up step as 200 and and the total step as
2000. For FairFace and CelebA data sets, we adopt the PyTorch (Paszke et al., 2017) version of ResNet-18
(He et al., 2016) as the feature extractor. We set the learning rate at 0.0002 without any learning rate
scheduling and set the final layer's hidden dimension to 128. We use batch training with a batch size of 256.
For all the data sets, we do five independent runs with random seeds [40, 41, 42, 43, 44]. We set the relative
preference of our method as [1, 5, 20, 20] for all data sets. We report the test results of the model which
achieves the best trade-off of task performance and fairness on the validation set. In our implementation,
for the convenience of computation of gradients, we use BackPACK (Dangel et al., 2020) package built on
PyTorch, and use Gurobi (Gurobi Optimization, 2021) with an academic license for the EPO solver. For the
experiments of three regularization terms separately, we search within the range of $[0.1, 1, 10, 100, 1e3, 1e4]$
to identify the parameter for the best fairness and task performance trade-off on the validation set. We run
all the experiments with four Tesla V100 SXM2 GPUs and a 12-core 2.2GHz CPU.

## F    Additional Experiments

### F.1    Sensitivity Analysis of Preference Vector

A core design of FAIRGM is the use of preference-aware multi-objective optimization to balance task per-
formance and multiple fairness regularization terms. In this subsection, we study how the choice of the
preference vector $\mathbf{p} = [p_E, p_0, p_1, p_2]$ affects the final fairness–accuracy trade-off.

We systematically vary the relative preference vector on two representative datasets (COMPAS and Fair-
Face), sweeping over six configurations: $[1, 1, 1, 1]$, $[1, 5, 5, 5]$, $[1, 5, 20, 20]$, $[1, 10, 50, 50]$, $[1, 10, 50, 100]$, and
$[1, 20, 100, 100]$. All other experimental settings are kept identical to Section 5.3. The results are reported
in Table 7.

Several observations can be drawn from Table 7. First, all preference configurations improve fairness over
ERM, confirming that the MOO formulation is effective regardless of the specific preference choice. Second,
the fairness–accuracy trade-off shifts gradually as the relative preference for fairness increases: higher fairness
preferences generally lead to lower DA and DEO at the cost of modest accuracy reductions. For example,
on COMPAS, DEO decreases from 0.217 under $[1, 1, 1, 1]$ to 0.109 under $[1, 20, 100, 100]$, while accuracy
decreases only from 0.671 to 0.669. On FairFace, DEO decreases from 0.034 to 0.008 with accuracy remaining
within 0.860–0.866 across all configurations. Third, the default preference $[1, 5, 20, 20]$ consistently achieves
a favorable trade-off on both datasets: it attains the best or near-best DA and competitive DEO while
maintaining high accuracy and F1. Notably, this single default is used across all datasets in our experiments
without per-dataset tuning.

**Practical guidance.**    At the Pareto-optimal solution, the EPO solver enforces $p_i \mathcal{L}_i \approx p_j \mathcal{L}_j$ for all pairs
of objectives $i, j$. This implies that the ideal preference is inversely proportional to the target loss values,

Table 7: Preference sensitivity analysis on COMPAS and FairFace datasets. The preference vector $\mathbf{p} = [p_E, p_0, p_1, p_2]$ controls the relative importance of ERM loss and the three fairness regularizations. All experiments use TPR as the fairness metric.

| Preference $\mathbf{p}$ | COMPAS | | | | FairFace | | | |
|---|---|---|---|---|---|---|---|---|
| | Acc ↑ | F1 ↑ | DA ↓ | DEO ↓ | Acc ↑ | F1 ↑ | DA ↓ | DEO ↓ |
| ERM (no fairness) | 0.675 | 0.620 | 0.027 | 0.298 | 0.867 | 0.843 | 0.072 | 0.061 |
| [1, 1, 1, 1] | 0.671 | 0.618 | 0.019 | 0.217 | 0.865 | 0.842 | 0.057 | 0.034 |
| [1, 5, 5, 5] | 0.673 | 0.626 | 0.015 | 0.184 | 0.863 | 0.840 | 0.049 | 0.018 |
| [1, 5, 20, 20] | 0.676 | 0.641 | 0.006 | 0.123 | 0.866 | 0.847 | 0.040 | 0.009 |
| [1, 10, 50, 50] | 0.677 | 0.637 | 0.009 | 0.140 | 0.863 | 0.846 | 0.044 | 0.008 |
| [1, 10, 50, 100] | 0.672 | 0.629 | 0.005 | 0.118 | 0.862 | 0.842 | 0.040 | 0.010 |
| [1, 20, 100, 100] | 0.669 | 0.624 | 0.014 | 0.109 | 0.860 | 0.836 | 0.042 | 0.008 |

Table 8: Comparison of last-layer fine-tuning with full tuning on Adult data set.

| Models | Acc ↑ | F1 ↑ | DA ↓ | DEO ↓ |
|---|---|---|---|---|
| ERM | $0.843_{\pm 0.015}$ | $0.698_{\pm 0.029}$ | $0.061_{\pm 0.009}$ | $0.042_{\pm 0.014}$ |
| Reduction | $0.808_{\pm 0.021}$ | $0.691_{\pm 0.017}$ | $0.042_{\pm 0.011}$ | $0.033_{\pm 0.012}$ |
| Adv | $0.839_{\pm 0.013}$ | $0.696_{\pm 0.014}$ | $0.047_{\pm 0.008}$ | $0.025_{\pm 0.015}$ |
| FairGrad | $0.832_{\pm 0.011}$ | $0.695_{\pm 0.012}$ | $0.014_{\pm 0.010}$ | $0.018_{\pm 0.012}$ |
| FAIRGM$_0$ | $0.828_{\pm 0.017}$ | $0.693_{\pm 0.018}$ | $0.021_{\pm 0.011}$ | $0.011_{\pm 0.009}$ |
| FAIRGM$_1$ | $0.837_{\pm 0.007}$ | $0.697_{\pm 0.010}$ | $0.016_{\pm 0.008}$ | $0.015_{\pm 0.009}$ |
| FAIRGM$_2$ | $0.839_{\pm 0.013}$ | $0.702_{\pm 0.011}$ | $0.013_{\pm 0.007}$ | $0.017_{\pm 0.010}$ |
| FAIRGM | $0.840_{\pm 0.009}$ | $0.704_{\pm 0.013}$ | $0.014_{\pm 0.009}$ | $0.015_{\pm 0.011}$ |
| FAIRGM$_{\text{full}}$ | $0.844_{\pm 0.012}$ | $0.706_{\pm 0.018}$ | $0.015_{\pm 0.007}$ | $0.013_{\pm 0.010}$ |

i.e., $\mathbf{p}^* \propto (\epsilon_E^{-1}, \epsilon_0^{-1}, \epsilon_1^{-1}, \epsilon_2^{-1})^\top$. In practice, practitioners can estimate the relative magnitudes of the ERM loss and fairness regularization values from a short warm-up run, and set the preference vector accordingly. Assigning relatively larger preferences to the first- and second-moment regularizations (e.g., $p_1 = p_2 = 20$ vs. $p_0 = 5$) reflects the empirical observation that gradient-level alignment terms tend to have smaller loss magnitudes than the zeroth-moment metric disparity, and thus require higher preference to be effectively optimized.

## F.2  Effect of Last-Layer Fine-Tuning on Adult Dataset

In Section 5.5, we demonstrated on the CI-MNIST benchmark that last-layer fine-tuning achieves comparable performance to full model training. To verify that this finding extends to the multi-group setting, we compare FAIRGM (last-layer fine-tuning) with FAIRGM$_{\text{full}}$ (all parameters updated) on the Adult dataset with four intersectional sensitive groups. The results are reported in Table 8.

The two variants perform comparably across all metrics. FAIRGM$_{\text{full}}$ achieves slightly higher accuracy and lower DEO, while FAIRGM achieves slightly better DA. The differences are within the standard error across runs, indicating that the fairness improvements are largely attributable to the final classifier adjustment rather than representation changes. This is consistent with the findings in Section 5.5 and prior work on last-layer retraining (Kirichenko et al., 2022; Mao et al., 2023), and confirms that last-layer fine-tuning remains an effective and computationally efficient strategy for FAIRGM in the multi-group setting.

## F.3  Trade-off between the task loss and the three regularizations (RQ2)

In this subsection, we use a toy example to compare the three moments of regularization. Following Donini et al. (2018), we generate a synthetic binary classification data set with two sensitive groups. For each group in class -1 and for the group $a$ in class 1, we generate 1,000 examples for training and the same number for

testing. For the group $b$ in class 1, we generate 200 examples for training and the same number for testing. Each set of examples is sampled from a 2-dimensional isotropic Gaussian distribution with different mean $\mu$ and variance $\sigma^2$ : (i) Group $a$, Label 1: $\mu = $ (-1, -1), $\sigma^2 = 0.8$; (ii) Group $a$, Label -1: $\mu = $ (1, 1), $\sigma^2 = 0.8$; (iii) Group $b$, Label 1: $\mu = $ (-0.5, -0.5), $\sigma^2 = 0.5$; (iv) Group $b$, Label -1: $\mu = $ (0.5, 0.5), $\sigma^2 = 0.5$. We use a linear classifier as the base model. When a standard machine learning method is applied to this synthetic data set, the generated model is unfair with respect to the group $b$, in that the classifier tends to negatively classify the examples in this group. We search in $[0.1, 1, 10, 100]$ for the best regularization weight for $\textsc{FairGM}_0$, $\textsc{FairGM}_1$ and $\textsc{FairGM}_2$, which only use the zero-moment, first-moment and second-moment regularization respectively, and set the preference by default as [1, 5, 20, 20] for $\textsc{FairGM}$. On this data set, we use the TPR loss as the fairness metric loss, and we show the results of three independent runs in Table 9. In the table, it is evident that the model exhibits improved fairness performance with zero-moment regularization, but at the great expense of task performance. First-moment regularization yields better task performance, but with limited gains in fairness. Second-moment regularization falls short of achieving the same level of fairness as zero-moment regularization but maintains task performance more effectively. For additional examples showcasing the benefits of first-moment and second-moment gradient matching, the works of Shi et al. (2022); Rame et al. (2022) serve as useful references. Note that these trends may vary across different data sets, as indicated in Tables 1 and 3. Yet, by synergizing the three regularizations, our model strikes a much better balance, achieving either comparable or superior task performance alongside enhanced fairness and robustness. In conclusion, each of the three regularizations presents distinct advantages and disadvantages, making it essential to allow users to determine their own preferences. This rationale drives our approach of integrating all three moments of regularization and utilizing the automated weighted training approach to cater to user preferences.

Table 9: Comparison of ERM and three regularizations on the synthetic data.

| Methods | Acc | F1 | DA | DEO |
|---------|-----|-----|-----|-----|
| ERM | $0.8813_{\pm 0.1851}$ | $0.8267_{\pm 0.1773}$ | $0.0500_{\pm 0.0232}$ | $0.0857_{\pm 0.0311}$ |
| $\textsc{FairGM}_0$ | $0.8240_{\pm 0.2031}$ | $0.7019_{\pm 0.2043}$ | $0.0417_{\pm 0.0143}$ | $0.0621_{\pm 0.0167}$ |
| $\textsc{FairGM}_1$ | $0.8819_{\pm 0.1622}$ | $0.8296_{\pm 0.1946}$ | $0.0507_{\pm 0.0128}$ | $0.0844_{\pm 0.0210}$ |
| $\textsc{FairGM}_2$ | $0.8692_{\pm 0.1877}$ | $0.8042_{\pm 0.1720}$ | $0.0378_{\pm 0.0217}$ | $0.0789_{\pm 0.0218}$ |
| $\textsc{FairGM}$ | $0.8814_{\pm 0.1140}$ | $0.8295_{\pm 0.1698}$ | $0.0402_{\pm 0.0133}$ | $0.0703_{\pm 0.0159}$ |

## G   Limitations and Broader Impacts

Limitations of this work include the restriction of our theoretical analysis to linear classifiers, sensitivity to batch size in small-group settings, and dependence on a user-specified preference vector, which we provide practical guidance for setting based on objective magnitudes at convergence. We hope our work on the preference-aware optimization framework towards fairness could provide increased flexibility to system designers when developing decision systems with consideration of fairness.

