# OpenReview forum: "Harmonizing Gradient Matching For Fairness"
_TMLR — Accepted by TMLR_

### Review · Reviewer_3Nwy · 2026-05-05

**Summary Of Contributions:**

This paper introduces Fair Gradient Matching (FairGM), an in-processing fairness framework that harmonizes group-conditioned optimization signals. FairGM aligns the gradient statistics of a fairness objective across groups at three levels: zeroth-moment (metric parity), first-moment (mean gradient alignment), and second-moment (gradient variance alignment). The training is formulated as a multi-objective optimization problem and solved via a preference-aware Pareto method (EPO). Empirical evaluations on synthetic and real-world datasets demonstrate superior fairness–accuracy trade-offs compared to existing baselines. The core contribution of this work lies in shifting the focus from solely optimizing fairness metrics to harmonizing gradient signals across groups.

**Audience:**

Yes

**Audience Explanation:**

This paper presents an approach to fairness that shifts the focus from achieving metric parity at a single solution to harmonizing gradient signals across groups, a perspective of potential interest to the broader research community.

**Broader Impact Concerns:**

None.

**Claims And Evidence:**

Yes

**Claims Explanation:**

Both the theoretical analysis and experimental results validate the use of first- and second-moment gradient matching.

**Requested Changes:**

1. In Figure 1, the authors claim that (c) second-moment gradient matching only ensures that the variances of the fairness metric are close. However, visually, the variances for Group 1 and Group 2 appear closer in Figure 1(b) than in Figure 1(c).
2. The Second-moment Regularization uses element-wise gradient variance, ignoring cross-parameter covariances, this may miss important structure in the optimization landscape. Could the authors provide a brief discussion on why covariances were not used?
3. While the authors claim that solely focusing on fairness metric balance may result in “unstable fairness behavior or sensitivity to perturbations” and state this as a motivation, there is no experiment to validate how the proposed FairGM is robust to perturbations.
4. This work only considers the binary group case. While the authors point this out in the Limitations section and provide brief guidance on how to generalize it to a multi-group scenario, I believe the extension to multiple groups should be included in the main paper as a major setting.

---

> ### Author Response · Authors · 2026-05-26
> **Response to Reviewer 3Nwy**
>
> We thank the reviewer for the thoughtful feedback. We address each point below.
>
> ` Comment 1: In Figure 1, the variances in (b) appear closer than in (c) `
>
> We have revised Figure 1 in the updated manuscript to make the distinction between sub-figures more visually apparent. The updated illustration shows that first-moment matching (b) aligns the slopes but leaves the variance bands different, while second-moment matching (c) aligns the variance bands but may leave the slopes different. Notes that each sub-figure illustrates one possible configuration and that visual similarity of variances in (b) should not be interpreted as a property of first-moment matching.
>
> ` Comment 2: Why element-wise gradient variance instead of the full covariance?`
>
> We chose element-wise variance over the full covariance matrix for three reasons:
>
> **Computational cost.** The full gradient covariance matrix has size $|\theta| \times |\theta|$, where $|\theta|$ is the number of model parameters. Even with last-layer fine-tuning, computing and matching full covariance matrices across groups would be substantially more expensive than matching element-wise variances, which requires only $O(|\theta|)$ computation per group.
>
> **Consistency with prior work.** Our design follows Rame et al. (2022), which uses element-wise gradient variance for domain generalization and provides analysis showing this is sufficient for promoting generalization across distributions. Our Proposition 4.1 shows that element-wise variance matching enforces the second moment of the feature-weighted prediction error distribution under $r + 1$ weighting schemes, indicating that it is a meaningful distributional constraint even without cross-parameter covariance.
>
> **Empirical sufficiency.** Our results across six datasets (including the newly added Adult and FairFace multi-group settings in Section 5.7) demonstrate that element-wise variance matching, combined with zeroth- and first-moment regularizations, achieves strong fairness improvements consistently. The marginal benefit of full covariance matching would need to justify the computational overhead.
>
> We have added a brief discussion of this design choice and the potential of structured covariance approximations (e.g., block-diagonal or low-rank) as future work in the revised manuscript.
>
> ` Comment 3: No experiment validates robustness to perturbations`
>
> We thank the reviewer for raising this point. We clarify that our use of "sensitivity to perturbations" refers to the behavior of the optimization process, not adversarial robustness. In practice, gradient-based optimizers do not converge to the exact optimum. The final parameters lie in a neighborhood around it. If the fairness landscape is asymmetric across groups in this neighborhood (i.e., the gradients and gradient variances differ), small parameter updates can reintroduce fairness disparities even when the fairness metric is balanced at the current iterate. This is the motivation for aligning first- and second-moment gradient statistics: they encourage the fairness landscape to behave similarly across groups in the neighborhood of the solution, not just at a single point.
>
> Our experimental results provide evidence for this effect. Compared to FairGM₀ (which only matches the fairness metric value at the solution), the full FairGM achieves lower variance across independent runs (Tables 1–2), indicating more consistent fairness outcomes under different optimization trajectories. The batch size study (Section 5.6) further shows that FairGM is robust to the noise introduced by stochastic gradient estimation. We have revised the manuscript to clarify that our stability argument concerns the optimization neighborhood, not adversarial perturbations.
>
> ` Comment 4: Extension to multiple groups should be in the main paper `
>
> In the revised manuscript, we have made two changes:
>
> **Multi-group discussion in the main text.** We have added a remark in Section 3.2 presenting the multi-group extension.
>
> **Multi-group experiments.** We added experiments on two datasets with more than two groups (Section 5.7 and Table 4). On the Adult dataset, we construct four intersectional groups from race and gender. On FairFace(Multi), we extend the sensitive attribute to three race groups (Black, White, Latino). FairGM achieves the best overall trade-off on both datasets. Notably, on FairFace(Multi), FairGM's DEO (0.009) matches the binary-group result, confirming that the method scales to multiple groups without fairness degradation. The ablation pattern (FairGM₀ best on DEO alone, FairGM best on overall trade-off) is consistent with the binary-group findings, demonstrating that the complementary roles of the three moments are preserved in the multi-group setting.

---

### Review · Reviewer_CXpa · 2026-05-07

**Summary Of Contributions:**

This paper addresses the problem of ensuring fairness in performance metrics when training a model using data composed of multiple demographic groups.
Typically, the zero-order method has been used, which directly regularizes the difference in fairness metrics between groups. However, models obtained using this method exhibit unstable behavior under perturbations.
Therefore, this paper attempts to solve this problem by proposing a method that uses gradient information for each group. Specific contributions are as follows:
* Proposal of gradient matching, which adds first- and second-order statistics to regularization, and a method for optimization within a multi-objective optimization framework.
* Attempting to provide theoretical guarantees for the usefulness of the method in setting up the training of a linear classifier using cross-entropy.
* Extensive numerical experiments verifying the usefulness of the proposed method.

Overall, this appears to be a solid contribution relevant to TMLR authorship, but there are some concerns that need revision before publication.

**Audience:**

Yes

**Audience Explanation:**

The problem of fairness is one of the central problem in modern machine learning. Therefore, the paper should attract at least some authorship.

**Claims And Evidence:**

No

**Claims Explanation:**

* Theorem 4.2 is misleading and appears incorrect as stated. The theorem says that for any $\epsilon,\delta>0$, a sufficient sample size implies that FairGM yields an $\epsilon$-approximated solution satisfying $|p_iL_i - p_jL_j|\le \epsilon$. However, Appendix A.2 later defines $\delta$ as a solution of an optimization problem. Hence $\delta$ is clearly not an arbitrary parameter. This point should be properly corrected in Theorem 4.2 of the main text.
* In Figure 3, the authors shows that the proposed method (fine-tuning the linear classifier with MOO after full-training) is compatible with the full-training. However, very high accuracy is achieved by all of the method, and the problem itself might be too easy to solve. I think a more difficult classification problem would be more appropriate to show the superiority of the proposed method. I recommend the authors to include at least one more difficult classification problem as a test bed.

**Requested Changes:**

See the comment regarding the claims section.

---

> ### Author Response · Authors · 2026-05-26
> **Response to Reviewer CXpa**
>
> We thank the reviewer for the careful reading and constructive suggestions. We address each point below.
>
> `  Comment 1: Theorem 4.2 is misleading — $\delta$ is not an arbitrary parameter `
>
> Thank you for pointing this out. We have corrected the theorem statement in the revised manuscript. Specifically:
>
> - $\delta$ is now explicitly defined within the theorem statement as the margin parameter.
> - The condition $\delta > 0$ is stated as a structural assumption, not a free parameter.
> - A remark in Appendix A.2 explains that $\delta$ measures how well-separated every hypothesis in $\mathcal{F}$ is from the $\epsilon$ -boundary of the preference constraint, and that for a finite $\mathcal{F}$, $\delta>0$ fails only for the finitely many values of $\epsilon$ at which some hypothesis lies exactly on the boundary, making the condition generic.
>
> The sample complexity bound, the proof structure, and all conclusions remain unaffected.
>
>
> ` Comment 2: Include a more difficult classification problem `
>
> We appreciate this suggestion. We have added experiments on two additional settings in Section 5.7 in the revised manuscript:
>
> **Adult dataset (multi-group).** We construct four disjoint sensitive groups by combining race (White vs. non-White) and gender (Male vs. Female). This is a larger dataset (~48K samples) with intersectional groups, which is a more challenging fairness setting than binary groups. As shown in Table 4, FairGM achieves the best accuracy (0.840) and F1 (0.704) while maintaining competitive fairness (DA: 0.014, DEO: 0.015).
>
> **FairFace (multi-group).** We extend the binary race attribute to three groups (Black, White, Latino), creating a multi-group fairness problem. FairGM achieves the best accuracy (0.860), F1 (0.833), and DEO (0.009), with DA (0.037) also among the lowest. Notably, the DEO matches the binary-group result, indicating the method scales to multiple groups without fairness degradation.
>
> **Last-layer vs. full tuning on Adult.** We also compare FairGM (last-layer fine-tuning) with FairGM_full (all parameters updated) on the Adult dataset (Appendix F.2 and Table 8). Given limited computation resources, we weren't able to compare on the FairFace data set. On the Adult data set, the two variants perform comparably. FairGM_full achieves slightly better accuracy (0.844 vs. 0.840) and DEO (0.013 vs. 0.015). This confirms that last-layer fine-tuning remains effective even on a harder, multi-group dataset.
>
> Note that our original evaluation already included four real-world datasets: CelebA (a large-scale image dataset with ~200K images and DEO of 0.498 under ERM, which is a genuinely difficult fairness problem), FairFace, COMPAS, and German. The CI-MNIST experiments serve a complementary purpose of providing a controlled evaluation under known bias types.

---

> > ### Comment · Reviewer_CXpa · 2026-06-02
> >
> > I have read the revised manuscript.
> >
> > * Now, the statement of Theorem 4.2 is clearer than before.
> > * Furthermore, I believe that additional experiments have made it clearer that the proposed method is more generally valid.
> >
> > Overall, I am satisfied with the corrections to my points, and I have no further questions or requests.

---

### Review · Reviewer_Nizk · 2026-05-12

**Summary Of Contributions:**

This paper studies the problem of group-level fairness optimization in fair machine learning. Its core argument is that existing methods typically minimize the final fairness metric gap directly, while paying less attention to the discrepancy in optimization signals across different sensitive groups during training. To address this issue, the paper proposes Fair Gradient Matching (FairGM), which aligns the zeroth-moment metric gap, first-moment mean gradient, and second-moment gradient variance of the fairness objective across groups through three regularization terms. Furthermore, the authors formulate the ERM loss and the three fairness regularization terms as a preference-aware multi-objective optimization problem and solve it using EPO. The experimental results show that FairGM can generally reduce inter-group fairness disparities while maintaining good predictive performance across multiple synthetic and real-world datasets.

**Audience:**

Yes

**Audience Explanation:**

Researchers working on fairness-aware optimization and gradient-based debiasing would likely find the idea of aligning group-conditioned fairness gradients useful, even if the current theoretical claims need refinement.

**Broader Impact Concerns:**

I do not have major broader-impact concerns.

**Claims And Evidence:**

No

**Claims Explanation:**

The experiments support that FairGM can be practically useful, but the stronger theoretical and generality claims are not fully justified by the provided evidence.

**Requested Changes:**

1.  **The theoretical conclusion of Theorem 4.2 relies on strong assumptions, and its practical meaning is overstated.** Section 4.4 claims that Theorem 4.2 guarantees that *FairGM yields an ε-approximated solution of the fair model,* and the subsequent remark further states that FairGM can *reliably find* a solution satisfying the fairness conditions. However, the proof in Appendix A/B mainly provides a uniform convergence / sample complexity analysis between an empirical constrained problem and its population counterpart under a finite hypothesis class. It does not establish optimization convergence for the FairGM/EPO algorithm itself, nor does it cover non-convex neural network training, last-layer fine-tuning, or preference misspecification. The proof also relies on strong assumptions such as a finite hypothesis class, bounded losses, and an exact constrained optimizer. In particular, Assumption B.3 requires FairReg$_i$ to be expressible as the expectation of a bounded per-sample loss, whereas FairReg2 is a group-wise per-sample gradient variance and does not directly match a simple IID per-sample expectation form.
2.  **The paper lacks sensitivity analysis for the preference vector.** A core design of FairGM is the use of preference-aware MOO to balance task performance and multiple fairness regularization terms. However, the paper does not sufficiently analyze how the choice of the preference vector affects the final fairness-accuracy trade-off.
3.  **The paper contains several noticeable overclaims, including but not limited to the following:**
    -   Theorem 4.2 is presented as an algorithmic guarantee for FairGM. However, the proof does not cover EPO convergence, SGD training, non-convex neural networks, the two-stage training procedure, or optimization error; it is essentially a generalization-type argument under a finite hypothesis class.
    -   Proposition 4.1 and Theorem 4.2 are collectively described as an *end-to-end theoretical foundation.* Proposition 4.1 only applies to linear classifiers with cross-entropy loss, while Theorem 4.2 relies on strong assumptions. Therefore, they are insufficient to support general deep models, TPR surrogates, and the complete FairGM pipeline.
    -   The paper claims that preference-aware MOO can automatically determine the fairness-performance trade-off. In practice, however, the trade-off is still controlled by a user-specified preference vector, and the paper neither proves nor implements a mechanism for automatically selecting an appropriate preference.
4.   **How is the surrogate for non-differentiable fairness metrics defined?** Section 3.2.1 mentions that reparameterization tricks or surrogate functions can be used, and Section 5.1 states that the TPR loss follows Wang et al. (2024). However, the paper does not provide the specific surrogate formula, temperature/threshold settings, gradient stabilization strategy, or its relationship to the final DEO evaluation. Since FairGM’s gradient matching is directly based on the gradient of the fairness metric, the exact form of this surrogate can significantly affect the behavior of the method. I suggest that the authors provide the relevant definitions and implementation details in the main text or appendix.
5.   Proposition 4.1(ii) claims that if FairReg1 = FairReg2 = 0, then the second moments of the two groups are equal: $\frac{1}{n_0} \sum_i z_{0,i}^2 = \frac{1}{n_1} \sum_i z_{1,i}^2$. However, this conclusion appears to hold only under the condition $n_0 = n_1$, which the authors seem to have overlooked.

---

> ### Author Response · Authors · 2026-05-26
> **Response to Reviewer Nizk (1/2)**
>
> We thank the reviewer for the thorough and constructive feedback. We respond to each point below.
>
>
> ` Comment 1: Theorem 4.2 relies on strong assumptions and its practical meaning is overstated`
>
> Thank you for your feedback. In the revised manuscript, we have made the following corrections:
>
> **Scope of the theorem.** We have rewritten the remark following Theorem 4.2 to clearly state that it provides a *statistical guarantee*: with sufficient training data, the empirical preference-constrained problem and the population-level problem share the same feasible set and near-optimal ERM loss. The theorem does not cover EPO convergence, SGD training, non-convex optimization, last-layer fine-tuning, or preference misspecification. We have removed the phrases "reliably find" and "end-to-end theoretical foundation," and we now note that the convergence of the EPO solver is established separately in Mahapatra & Rajan (2020).
>
> **Finite hypothesis class.** This assumption is standard in sample complexity analyses (Shalev-Shwartz & Ben-David, 2014). Note that our last-layer fine-tuning strategy, where only the final linear classifier is updated atop a fixed feature extractor, makes this assumption substantially more appropriate than it would be for a general deep network.
>
> **Assumption B.3 and FairReg₂.** We have replaced Assumption B.3 with a corrected version (Now Assumption A.3) that handles each regularization appropriately:
>
> - Conditions (i) and (ii) are unchanged: the ERM loss and FairReg₀, FairReg₁ are bounded per-sample averages.
> - A new condition (iii) requires that per-sample gradients of the fairness metric are bounded: $\|g_s^i\| \leq G_{\max} < \infty$.
>
> This implies the U-statistic kernel $h(g^i, g^j) = (g^i - g^j)^2$ is bounded by $(2G_{\max})^2$. In the proof, we now apply Hoeffding's inequality for U-statistics (Hoeffding, 1963; Serfling, 2009) to FairReg₂, which yields the same exponential concentration rate as standard Hoeffding for the other losses. The proof structure is unchanged; only the concentration tool for FairReg₂ differs, and $L_{\max}$ is redefined as $\max(L_{\text{ERM}}, L^0_{\text{FairReg}}, L^1_{\text{FairReg}}, (2G_{\max})^2)$.
>
> ` Comment 2: The paper lacks sensitivity analysis for the preference vector`
>
> We have added a sensitivity study in the revised paper in Appendix F.1. Specifically:
> - We vary the preference vector on representative datasets (COMPAS and FairFace), sweeping over different ratios between the ERM preference and fairness regularization preferences (e.g., [1, 1, 1, 1], [1, 5, 5, 5], [1, 5, 20, 20], [1, 10, 50, 50], [1, 10, 50, 100], [1, 20, 100, 100]), reporting both accuracy and fairness metrics.
> - The results demonstrate that FairGM's performance degrades gracefully across a reasonable range of preferences, confirming that the framework is not highly sensitive to the exact preference choice.
> - We provide practical guidance: since the EPO solver enforces $p_i \mathcal{L}_i = p_j \mathcal{L}_j$ at optimality, the ideal preference is inversely proportional to the target loss values. Practitioners can estimate these from a short ERM warm-up run and set preferences accordingly.
> We also note that the default preference [1, 5, 20, 20] is used across all datasets in our experiments without per-dataset tuning.
>
>
> ` Comment 3: The paper contains several noticeable overclaims`
>
> We have corrected these overclaims in the revised manuscript.
>
> **3a. Theorem 4.2 as an "algorithmic guarantee."** As discussed above, Theorem 4.2 is now labeled as a sample complexity result. The revised remark states: "Theorem 4.2 provides a sample complexity guarantee ensuring that the empirical preference-constrained problem approximates the population-level problem." Optimization convergence of EPO is cited separately.
>
> **3b. "End-to-end theoretical foundation."** The revised text reads: "Proposition 4.1 provides analytical justification for the design of each regularization term in the tractable linear-classifier setting, while Theorem 4.2 provides a complementary statistical guarantee." Together, they offer theoretical motivation rather than a complete end-to-end proof for the general deep learning pipeline. We also explicitly state the gap: extending Proposition 4.1 to non-linear models and surrogate losses remains an open theoretical question, though our extensive empirical results demonstrate practical effectiveness beyond the linear case.

---

> > ### Author Response · Authors · 2026-05-26
> > **Response to Reviewer Nizk (2/2)**
> >
> > **3c. Preference-aware MOO "automatically determines" the trade-off.** The revised text in the Introduction section now reads: "Given a user-specified preference vector, the preference-aware MOO solver automatically determines the descent direction that leads to the corresponding Pareto-optimal solution, eliminating the need for exhaustive hyperparameter search over regularization weights." This reflects that the EPO solver automates the optimization mechanics given $\mathbf{p}$, but the trade-off itself is controlled by the user's choice of $\mathbf{p}$. We note that automatic preference selection is an interesting direction for future work.
> >
> >
> > `Comment 4: How is the surrogate for non-differentiable fairness metrics defined?`
> >
> > We have added the complete surrogate definition to Appendix C. Specifically, we use a sigmoid-based surrogate to approximate the indicator function in the true positive rate. For group $s$:
> >
> > $\text{TPR}_s^{\text{surr}}(f) = \frac{\sum_{i: y_s^i = 1} \sigma(\tau \cdot f_\theta(x_s^i))}{\sum_{i: y_s^i = 1} 1}$
> >
> > where $\sigma(\cdot)$ is the sigmoid function and $\tau > 0$ is a temperature parameter. As $\tau \to \infty$, the surrogate converges to the true TPR. We set $\tau = 2$ in our implementation.
> >
> > **Relationship to DEO evaluation.** At evaluation time, DEO is computed using the hard indicator function (actual difference in true positive rates with a 0.5 threshold). The surrogate serves as a differentiable proxy during training only.
> >
> > ` Comment 5: Proposition 4.1(ii) requires $n_0 = n_1$`
> >
> > Thank you for pointing this out. We have fixed this by changing the definition of FairReg₂ to use population variance (dividing by $n_s$) instead of sample variance (dividing by $n_s - 1$):
> >
> > $$v_s = \frac{1}{n_s}\sum_{i=1}^{n_s}(g_s^i - \bar{g}_s)^2$$
> >
> > With this definition, FairReg₁ = FairReg₂ = 0 directly implies $\overline{z_0^2} = \overline{z_1^2}$ without any condition on group sizes. In practice, the difference between dividing by $n_s$ vs. $n_s - 1$ is negligible for the sample sizes in our experiments, so the experimental results are unaffected. A remark noting this change has been added to the revised paper.

---

> > > ### Comment · Reviewer_Nizk · 2026-05-31
> > >
> > > I have read the authors’ response and the revised manuscript. The authors have satisfactorily addressed my main concerns.
> > >
> > > In particular, the manuscript now more clearly explains the role of the preference vector in the preference-aware MOO framework, which resolves my concern that the original wording could be interpreted as automatically determining the fairness–performance trade-off. The added discussion of the TPR surrogate also provides the key implementation details needed to understand how the non-differentiable fairness metric is optimized in practice. In addition, the revision of FairReg2 and the corresponding clarification around Proposition 4.1 address my concern regarding the unequal-group-size case and strengthen the theoretical presentation.
> > >
> > > Overall, I am satisfied with the revision and do not have further questions for the authors.

---

### Author Response · Authors · 2026-05-26
**Summary of Revisions**

We thank all reviewers for their detailed and constructive feedback, which has substantially improved the paper. Below we summarize the key modifications made in the revised manuscript.

**Theoretical Corrections**

We have revised the theoretical presentation of Theorem 4.2, Assumption B.3, and Proposition 4.1.

Theorem 4.2 now explicitly defines the margin parameter $\delta$ as a problem-dependent quantity, states $\delta > 0$ as a structural assumption, and is clearly labeled as a sample complexity / generalization result, not an algorithmic guarantee or an "end-to-end theoretical foundation." The remark honestly states that the theorem guarantees generalization for any fixed preference but not solution quality under preference misspecification, and notes that EPO convergence is established separately (Mahapatra & Rajan, 2020).

Assumption B.3 (Now Assumption A.3) has been corrected to handle FairReg₂ (gradient variance), which is not a per-sample average: the revised assumption adds a bounded per-sample gradient condition, and the proof now applies Hoeffding's inequality for U-statistics (Hoeffding, 1963; Serfling, 2009).

Proposition 4.1(ii) has been fixed by using population variance (dividing by $n_s$) in the definition of FairReg₂, eliminating the implicit $n_0 = n_1$ requirement.

Finally, the claim that preference-aware MOO "automatically determines the trade-off" has been revised to state that it "automatically determines the descent direction given a user-specified preference."



**New Experiments**

**Preference sensitivity analysis (Reviewer Nizk).** We systematically vary the preference vector on COMPAS and FairFace across six configurations. The results (Table 7) show that performance degrades gracefully across a reasonable range and that the default [1, 5, 20, 20] achieves a consistently favorable trade-off without per-dataset tuning.

**Multi-group experiments (Reviewers 3Nwy).** We add experiments with more than two sensitive groups on two datasets (Table 4): Adult (four intersectional groups from race × gender) and FairFace(Multi) (three race groups: Black, White, Latino). FairGM achieves the best overall trade-off on both datasets, and the ablation pattern is consistent with the binary-group findings.

**Last-layer vs. full tuning on Adult (Reviewer CXpa).** We compare FairGM with FairGM$_{full}$ on the Adult dataset (Table 8). The two variants perform comparably, confirming that last-layer fine-tuning remains effective in the multi-group setting.

---

### Decision · Action_Editor_Aemo · 2026-06-15

**Recommendation:** Accept as is

**Additional Comments:**

The authors should ensure that the final manuscript clearly preserves the revised scope of the theoretical claims, especially the distinction between the sample-complexity/generalization result and algorithmic convergence of the practical training procedure. They should also ensure that all newly added experiments, surrogate definitions, and multi-group extensions are clearly described and properly integrated into the final version.

**Audience:**

Yes

**Audience Explanation:**

The paper addresses fairness-aware optimization, group-conditioned training dynamics, and multi-objective learning, which are relevant to researchers working on trustworthy machine learning, algorithmic fairness, robust optimization, and fairness-aware model training. The proposed gradient-matching perspective is likely to be of interest to at least part of the TMLR audience.

**Claims And Evidence:**

Yes

**Claims Explanation:**

The submission studies group-level fairness optimization and proposes Fair Gradient Matching (FairGM), an in-processing fairness framework that aligns group-conditioned optimization signals. The method introduces zeroth-moment metric matching, first-moment gradient alignment, and second-moment gradient-variance alignment, and formulates the resulting training objective as a preference-aware multi-objective optimization problem solved with EPO.

* Reviewers were generally positive about the relevance of the problem and the usefulness of the proposed method. They agreed that fairness-aware optimization is an important topic for TMLR, and that the paper offers a meaningful perspective by moving beyond matching final fairness metrics toward aligning the optimization behavior across sensitive groups.

* The initial reviews raised several substantive concerns, including the scope and presentation of Theorem 4.2, the treatment of FairReg₂ under unequal group sizes, the lack of preference-vector sensitivity analysis, insufficient detail on the differentiable surrogate for non-differentiable fairness metrics, the absence of multi-group experiments, and the need to clarify the motivation for using element-wise gradient variance rather than full covariance.

In response to the reviews, the authors made substantial revisions:

- Theorem 4.2 was clarified as a sample-complexity/generalization result rather than an algorithmic convergence guarantee for the full FairGM/EPO training pipeline.
- The statement of Theorem 4.2 was revised by explicitly defining the margin parameter and presenting the corresponding condition as a structural assumption.
- The definition and analysis of FairReg₂ were corrected, including the unequal-group-size issue, and the corresponding assumptions and proof arguments were revised.
- A preference-sensitivity analysis was added to show how different preference vectors affect the fairness-accuracy trade-off.
The differentiable surrogate definition for the TPR-based fairness metric was provided, and its relationship to the final DEO evaluation was clarified.
- Multi-group experiments on Adult and FairFace, including intersectional groups and multiple race groups, were added to better support the generality of the method.
-The discussion of element-wise gradient variance was revised, and justification was added for not matching full covariance due to computational cost and empirical sufficiency.

After revision, the reviewers’ final recommendations were positive. Reviewers Nizk and CXpa explicitly stated that their main concerns were satisfactorily addressed. Reviewer Nizk noted that the revised manuscript resolved the concerns about the preference vector, the surrogate definition, and the unequal-group-size issue. Reviewer CXpa found the theorem statement clearer and the additional experiments helpful in demonstrating the broader validity of the method. Reviewer 3Nwy recommended Leaning Accept. Overall, the revised paper provides adequate evidence for its main claims.

---

> ### Author Response · Authors · 2026-07-13
>
> We thank the Action Editor for the positive decision and the helpful guidance for the final manuscript. We confirm that all requested items are properly addressed in the camera-ready version:
>
> **Theoretical claims.** The remark following Theorem 4.2 clearly states that it is a sample complexity result, and explicitly notes that optimization convergence of the EPO solver is a separate property established in Mahapatra & Rajan (2020). The phrases "reliably find" and "end-to-end theoretical foundation" have been replaced throughout with precisely scoped language.
>
> **New experiments.** The preference sensitivity analysis (Appendix F.1, Table 7), multi-group experiments on Adult and FairFace(Multi) (Section 5.7, Table 4), and last-layer vs. full tuning comparison on Adult (Appendix F.2, Table 8) are fully described with experimental setup, dataset construction details, and discussion of results.
>
> **Surrogate definition.** The sigmoid-based TPR surrogate formula, temperature setting, and its relationship to true TPR are in Appendix C, referenced from Section 5.1.
>
> **Multi-group extension.** A remark in Section 3.2 presents the pairwise generalization of all three regularizations to $K > 2$ groups, and the multi-group experiments in Section 5.7 demonstrate its effectiveness.
>
> We have carefully reviewed the final manuscript to ensure consistency between the main text, appendix, and all revised claims. Thank you again for the constructive feedback and support!